# Towards Empirical Sandwich Bounds on the Rate-Distortion Function

**Yibo Yang, Stephan Mandt**
Department of Computer Science, UC Irvine
`{yibo.yang,mandt}@uci.edu`

## Abstract

Rate-distortion (R-D) function, a key quantity in information theory, characterizes the fundamental limit of how much a data source can be compressed subject to a fidelity criterion, by *any* compression algorithm. As researchers push for ever-improving compression performance, establishing the R-D function of a given data source is not only of scientific interest, but also reveals the possible room for improvement in compression algorithms. Previous work on this problem relied on distributional assumptions on the data source (Gibson, 2017) or only applied to discrete data. By contrast, this paper makes the first attempt at an algorithm for sandwiching the R-D function of a general (not necessarily discrete) source requiring only i.i.d. data samples. We estimate R-D sandwich bounds for a variety of artificial and real-world data sources, in settings far beyond the feasibility of any known method, and shed light on the optimality of neural data compression (Ballé et al., 2021; Yang et al., 2022). Our R-D upper bound on natural images indicates theoretical room for improving state-of-the-art image compression methods by at least one dB in PSNR at various bitrates. Our data and code can be found here.

## 1 Introduction

From storing astronomical images captured by the Hubble telescope, to delivering familiar faces and voices over video calls, data compression, i.e., communication of the "same" information but with less bits, is commonplace and indispensable to our digital life, and even arguably lies at the heart of intelligence (Mahoney, 2009). While for lossless compression, there exist practical algorithms that can compress any discrete data arbitrarily close to the information theory limit (Ziv & Lempel, 1977; Witten et al., 1987), no such universal algorithm has been found for *lossy* data compression (Berger & Gibson, 1998), and significant research efforts have dedicated to lossy compression algorithms for various data. Recently, deep learning has shown promise for learning lossy compressors from raw data examples, with continually improving compression performance often matching or exceeding traditionally engineered methods (Minnen et al., 2018; Agustsson et al., 2020; Yang et al., 2020a).

However, there are fundamental limits to the performance of any lossy compression algorithm, due to the inevitable trade-off between *rate*, the average number of bits needed to represent the data, and the *distortion* incurred by lossy representations. This trade-off is formally described by the rate-distortion (R-D) function, for a given *source* (i.e., the data distribution of interest; referred to as such in information theory) and distortion metric. The R-D function characterizes the best theoretically achievable rate-distortion performance by any compression algorithm, which can be seen as a lossy-compression counterpart and generalization of Shannon entropy in lossless compression.

Despite its fundamental importance, the R-D function is generally unknown analytically, and establishing it for general data sources, especially real world data, is a difficult problem (Gibson, 2017). The default method for computing R-D functions, the Blahut-Arimoto algorithm (Blahut, 1972; Arimoto, 1972), only works for discrete data with a known probability mass function and has a complexity exponential in the data dimensionality. Applying it to an unknown data source requires discretization (if it is continuous) and estimating the source probabilities by a histogram, both of which introduce errors and are computationally infeasible beyond a couple of dimensions. Previous work characterizing the R-D function of images and videos (Hayes et al., 1970; Gibson, 2017) all assumed a statistical model of the source, making the results dependent on the modeling assumptions.

In this work, we make progress on this long-standing problem in information theory using tools from machine learning, and introduce new algorithms for upper and lower bounding the R-D function of a *general* (i.e., discrete, continuous, or neither), *unknown* memoryless source. More specifically,

1. Similarly to how a VAE with a discrete likelihood model minimizes an upper bound on the data entropy, we establish that *any $\beta$-VAE with a likelihood model induced by a distortion metric minimizes an upper bound on the data rate-distortion function*. We thus open the deep generative modeling toolbox to the estimation of an upper bound on the R-D function.

2. We derive a lower bound estimator of the R-D function that can be made asymptotically exact and optimized by stochastic gradient ascent. Facing the difficulty of the problem involving global optimization, we restrict to a squared error distortion for a practical implementation.

3. We perform extensive experiments and obtain non-trivial sandwich bounds on various data sources, including GAN-generated artificial sources and real-world data from speech and physics. Our results shed light on the effectiveness of neural compression approaches (Ballé et al., 2021; Minnen et al., 2018; Minnen & Singh, 2020), and identify the *intrinsic* (rather than nominal) dimension of data as a key factor affecting the tightness of our lower bound.

4. Our estimated R-D upper bounds on high-resolution natural images (evaluated on the standard Kodak and Tecnick datasets) indicate theoretical room for improvement of state-of-the-art image compression methods by at least one dB in PSNR, at various bitrates.

We begin by reviewing the prerequisite rate-distortion theory in Section 2, then describe our upper and lower bound algorithms in Section 3 and Section 4, respectively. We discuss related work in Section 5, report experimental results in Section 6, and conclude in Section 7.

## 2 BACKGROUND

Rate-distortion (R-D) theory deals with the fundamental trade-off between the average number of bits per sample (*rate*) used to represent a data source $X$ and the *distortion* incurred by the lossy representation $Y$. It asks the following question about the limit of lossy compression: for a given data source and a distortion metric (a.k.a., a fidelity criterion), what is the minimum number of bits (per sample) needed to represent the source at a tolerable level of distortion, regardless of the computation complexity of the compression procedure? The answer is given by the rate-distortion function $R(D)$. To introduce it, let the source and its reproduction take values in the sets $\mathcal{X}$ and $\mathcal{Y}$, conventionally called the *source* and *reproduction alphabets*, respectively. We define the data source formally by a random variable $X \in \mathcal{X}$ following a (usually unknown) distribution $P_X$, and assume a distortion metric $\rho : \mathcal{X} \times \mathcal{Y} \to [0, \infty)$ has been given, such as the squared error $\rho(x, y) = \|x - y\|^2$. The rate-distortion function is then defined by the following constrained optimization problem,

$$R(D) = \inf_{Q_{Y|X} : \, \mathbb{E}[\rho(X,Y)] \leq D} I(X; Y), \tag{1}$$

where we consider all random transforms $Q_{Y|X}$ whose expected distortion is within the given threshold $D \geq 0$, and minimize the mutual information between the source $X$ and its reproduced representation $Y$ [1]. Shannon's lossy source coding theorems (Shannon, 1948; 1959) gave operational significance to the above mathematical definition of $R(D)$, as the minimum achievable rate with which any lossy compression algorithm can code i.i.d. data samples at a distortion level within $D$.

The R-D function thus gives the tightest lower bound on the rate-distortion performance of any lossy compression algorithm, and can inform the design and analysis of such algorithms. If the operational distortion-rate performance of an algorithm lies high above the source $R(D)$-curve $(D, R(D))$, then further performance improvement may be expected; otherwise, its performance is already close to theoretically optimal, and we may focus our attention on other aspects of the algorithm. As the R-D function does not have an analytical form in general, we propose to estimate it from data samples, making the standard assumption that various expectations w.r.t. the true data distribution $P_X$ exist and can be approximated by sample averages. When the source alphabet is finite, this assumption automatically holds, and $R(D)$ also provides a lower bound on the Shannon entropy of discrete data.

---

[1] Both the expected distortion and mutual information terms are defined w.r.t. the joint distribution $P_X Q_{Y|X}$. We formally describe the general setting of the paper, including the technical definitions, in Appendix A.1.

## 3   UPPER BOUND ALGORITHM

R-D theory (Cover & Thomas, 2006) tells us that every (distortion, rate) pair lying above the $R(D)$-curve is in theory realizable by a (possibly expensive) compression algorithm. An upper bound on $R(D)$ thus reveals what kind of compression performance is theoretically achievable. Towards this goal, we borrow the variational principle of the Blahut-Arimoto (BA) algorithm, but extend it to a general (e.g., non-discrete) source requiring only its samples. Our resulting algorithm optimizes a $\beta$-VAE whose likelihood model is specified by the distortion metric, a common case being a Gaussian likelihood with a fixed variance. For the first time, we establish this class of models as computing a model-agnostic upper bound on the source R-D function, as defined by a data compression task.

**Variational Formulation.**   Following the BA algorithm (Blahut, 1972; Arimoto, 1972), we consider a Lagrangian relaxation of the constrained problem defining $R(D)$, which has the variational objective

$$\mathcal{L}(Q_{Y|X}, Q_Y, \lambda) := \mathbb{E}_{x \sim P_X}[KL(Q_{Y|X=x} \| Q_Y)] + \lambda \mathbb{E}_{P_X Q_{Y|X}}[\rho(X, Y)], \qquad (2)$$

where $Q_Y$ is a new, arbitrary probability measure on $\mathcal{Y}$. The first (*rate*) term is a variational upper bound on the mutual information $I(X; Y)$, and the second (*distortion*) term enforces the distortion tolerance constraint in Eq. 1. For each fixed $\lambda > 0$, the BA algorithm globally minimizes $\mathcal{L}$ w.r.t. the variational distributions $Q_{Y|X}$ and $Q_Y$ by coordinate descent; at convergence, the (distortion, rate) pair yields a point on the $R(D)$ curve (Csiszár, 1974a). Unfortunately, the BA algorithm only applies when $\mathcal{X}$ and $\mathcal{Y}$ are finite (hence discrete), and the source distribution known. Otherwise, a preprocessing step is required to discretize a continuous source and/or estimate source probabilities by a histogram, which introduces a non-negligible bias. This bias, along with its exponential complexity in the data dimension, also makes BA infeasible beyond a couple of (usually 2 or 3) dimensions.

**Proposed Method.**   To avoid these difficulties, we propose to apply (stochastic) gradient descent on $\mathcal{L}$ w.r.t. flexibly parameterized variational distributions $Q_{Y|X}$ and $Q_Y$. In this work we parameterize the distributions by neural networks, and predict the parameters of each $Q_{Y|X=x}$ by an *encoder* network $\phi(x)$ as in amortized inference (Kingma & Welling, 2014). Given data samples, the estimates of rate and distortion terms of $\mathcal{L}$ yield a point that in expectation lies on an R-D upper bound $R_U(D)$, and we tighten this bound by optimizing $\mathcal{L}$; repeating this procedure for various $\lambda$ traces out $R_U(D)$.

The objective $\mathcal{L}$ closely resembles the negative ELBO (NELBO) objective of a $\beta$-VAE (Higgins et al., 2017) if we view the reproduction space $\mathcal{Y}$ as the "latent space". The connection is immediate when $\mathcal{X}$ is continuous and a squared error $\rho$ specifies the density of a Gaussian likelihood $p(x|y) \propto \exp(-\|x-y\|^2)$. However, unlike in data compression, where $\mathcal{Y}$ is determined by the application (and often equal to $\mathcal{X}$ for a full-reference distortion), the latent space in a ($\beta$-)VAE typically has a lower dimension than $\mathcal{X}$, and a *decoder* network is used to parameterize a likelihood model in the data space. To capture this setup, we introduce a new, arbitrary latent space $\mathcal{Z}$ on which we define variational distributions $Q_{Z|X}, Q_Z$, and a (possibly stochastic) decoder function $\omega : \mathcal{Z} \to \mathcal{Y}$. This results in an extended objective, resembling a $\beta$-VAE with a likelihood density $p(x|z) \propto \exp\{-\rho(x, \omega(z)\}$,

$$J(Q_{Z|X}, Q_Z, \omega, \lambda) := \mathbb{E}_{x \sim P_X}[KL(Q_{Z|X=x} \| Q_Z)] + \lambda \mathbb{E}_{P_X Q_{Z|X}}[\rho(X, \omega(Z))]. \qquad (3)$$

This objective is closely related to the original data compression task and provides an upper bound on the source $R(D)$, as follows. Treating $\mathcal{Z}$ as the reproduction alphabet, we can define a new distortion $\rho_\omega(x, z) := \rho(x, \omega(z))$, and a $\omega$-induced R-D function, $R_\omega(D) := \inf_{Q_{Z|X}:\mathbb{E}[\rho_\omega(X,Z)] \le D} I(X; Z)$, for each choice of $\omega$. Our Theorem A.3 then guarantees that $R_\omega(D) \ge R(D)$, for any $\omega$, and consequently the (distortion, rate) of $J$ always lies above $R(D)$. Moreover, $R_\omega(D) = R(D)$ for a bijective $\omega$, which offers some theoretical support for the use of invertible pixel-shuffle operations instead of upsampled convolutions in the decoder of image compression autoencoders (Theis et al., 2017; Cheng et al., 2020). We can now minimize the NELBO-like objective $J$ w.r.t. the parameters of $(Q_{Z|X}, Q_Z, \omega)$ similar to training a $\beta$-VAE, knowing that we are optimizing an upper bound on the rate-distortion function of the source. This can be seen as the lossy counterpart to the lossless setting, where it is well-established that minimizing the NELBO minimizes an upper bound on the Shannon entropy of the source (Frey & Hinton, 1997; MacKay, 2003), the limit of lossless data compression.

The extended objective offers the freedom to define variational distributions on any suitable latent space $\mathcal{Z}$, rather than $\mathcal{Y}$, which we found to simplify the modeling task and yield tighter bounds. E.g., even if $\mathcal{Y}$ is high-dimensional and discrete, we can still work with densities on a continuous and

lower-dimensional $\mathcal{Z}$ and draw upon tools such as normalizing flows (Kobyzev et al., 2021). We can also treat $Z$ as the concatenation of sub-vectors $[Z_1, Z_2, ..., Z_L]$, and parameterize $Q_Z$ in terms of simpler component distributions $Q_Z = \prod_{l=1}^{L} Q_{Z_l|Z_{<l}}$ (similarly for $Q_{Z|X}$) as in a hierarchical VAE.

## 4 LOWER BOUND ALGORITHM

Without knowing the tightness of an R-D upper bound, we could be wasting time and resources trying to improve the R-D performance of a compression algorithm, when it is in fact already close to the theoretical limit. This would be avoided if we could find a matching *lower* bound on $R(D)$. Unfortunately, the problem turns out to be much more difficult computationally. Indeed, every compression algorithm, or every pair of variational distributions $(Q_Y, Q_{Y|X})$ yields a point above $R(D)$. Conversely, establishing a lower bound requires disproving the existence of *any* compression algorithm that can conceivably operate below the $R(D)$ curve. In this section, we derive an algorithm that can in theory produce arbitrarily tight R-D lower bounds. However, as an indication of its difficulty, the problem requires *globally* maximizing a family of partition functions. By restricting to a continuous reproduction alphabet and a squared error distortion, we make some progress on this problem and obtain useful lower bounds especially on data with low intrinsic dimension (see Sec. 6).

**Dual characterization of $R(D)$.** While upper bounds on $R(D)$ arise naturally out of its definition as a minimization problem, a variational lower bound requires expressing $R(D)$ through a *maximization* problem. For this, we introduce a "dual" function as the optimum of the Lagrangian Eq. 2 ($Q_Y$ is eliminated by replacing the rate upper bound with the exact mutual information $I(X; Y)$):

$$F(\lambda) := \inf_{Q_{Y|X}} I(X; Y) + \lambda \mathbb{E}[\rho(X, Y)]. \qquad (4)$$

As illustrated by Fig. 1, $F(\lambda)$ is the maximum $R$-axis intercept of a straight line with slope $-\lambda$, among all such lines that lie below or tangent to $R(D)$; the R-D curve can then be found by taking the upper envelope of lines with slope $-\lambda$ and R-intercept $F(\lambda)$, i.e., $R(D) = \max_{\lambda \geq 0} F(\lambda) - \lambda D$. This key result is captured mathematically by Lemma A.1, and the following theorem:

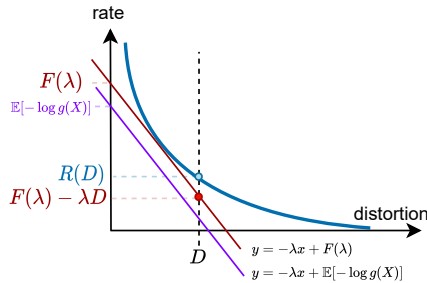

Figure 1: The geometry of the R-D lower bound problem. For a given slope $-\lambda$, we seek to maximize the $rate$-axis intercept, $\mathbb{E}[-\log g(X)]$, over all $g \geq 0$ functions admissible according to Eq. 6.

**Theorem 4.1.** *(Csiszár, 1974b)* (As follows, all the expectations are with respect to the data source r.v. $X \sim P_X$.) *Under basic conditions (e.g., satisfied by a bounded $\rho$; see Appendix A.2), it holds that*

$$F(\lambda) = \max_{g(x)} \{\mathbb{E}[-\log g(X)]\}, \qquad (5)$$

*where the maximization is over all non-negative functions $g : \mathcal{X} \to [0, \infty)$ satisfying the constraint*

$$\mathbb{E}\left[\frac{\exp(-\lambda \rho(X, y))}{g(X)}\right] = \int \frac{\exp(-\lambda \rho(x, y))}{g(x)} dP_X(x) \leq 1, \forall y \in \mathcal{Y}. \qquad (6)$$

In other words, every admissible $g$ yields a lower bound of $R(D)$, via an underestimator of the intercept $\mathbb{E}[-\log g(X)] \leq F(\lambda)$. We give the origin of this result in related work in Section 5.

**Proposed Unconstrained Formulation.** The constraint in Eq. 6 is concerning—it is a family of possibly infinitely many constraints, one for each $y$. To make the problem easier to work with, we propose to eliminate the constraints by the following transformation. Let $g$ be defined in terms of another function $u(x) \geq 0$ and a scalar $c$ depending on $u$, such that

$$g(x) := cu(x), \quad \text{where} \quad c := \sup_{y \in \mathcal{Y}} \Psi_u(y), \quad \text{and} \quad \Psi_u(y) := \mathbb{E}\left[\frac{\exp -\lambda \rho(X, y)}{u(X)}\right]. \qquad (7)$$

This reparameterization of $g$ is without loss of generality, and can be shown to always satisfy the constraint in Eq. 6. While this form of $g$ bears a superficial resemblance to an energy-based model

(LeCun et al., 2006), with $\frac{1}{c}$ resembling a normalizing constant, there is an important difference: $c = \sup_y \Psi_u(y)$ is in fact the supremum of a family of "partition functions" $\Psi_u(y)$ indexed by $y$; we thus refer to $c$ as the *sup-partition function*. Although all these quantities have $\lambda$-dependence, we omit this from our notation since $\lambda$ is a fixed input parameter (as in the upper bound algorithm).

Consequently, $F$ is now the result of *unconstrained* maximization over all $u$ functions, and we obtain a lower bound on it by restricting $u$ to a subset of functions with parameters $\theta$ (e.g., neural networks),

$$F(\lambda) = \max_{u \geq 0}\{\mathbb{E}[-\log u(X)] - \log \sup_{y \in \mathcal{Y}} \Psi_u(y)\} \geq \max_{\theta}\{\mathbb{E}[-\log u_\theta(X)] - \log \sup_{y \in \mathcal{Y}} \Psi_\theta(y)\}$$

Define the $\theta$-parameterized objective $\ell(\theta) := \mathbb{E}[-\log u_\theta(X)] - \log c(\theta)$, with $c(\theta) = \sup_{y \in \mathcal{Y}} \Psi_\theta(y)$. Given samples of $X$, we can in principle maximize $\ell(\theta)$ by (stochastic) gradient ascent. However, computing the sup-partition function $c(\theta)$ poses serious computation challenges: even evaluating $\Psi_\theta(y)$ for a single $y$ value involves a potentially high-dimensional integral w.r.t. $P_X$; this is only exacerbated by the need to globally optimize w.r.t. $y$, an NP-hard problem even in one-dimension.

**Proposed Method.** To tackle this problem, we propose an over-estimator of the sup-partition function inspired by IWAE (Burda et al., 2015). Fix $\theta$ for now; we denote the integrand in Eq. 7 by $\psi(x, y) := \frac{\exp{-\lambda \rho(x,y)}}{u(x)}$ (so $c = \sup_{y \in \mathcal{Y}} \mathbb{E}[\psi(X, y)]$), and omit the dependence on $\theta$ to simplify notation. Given $k \geq 1$ i.i.d. random variables $X_1, ..., X_k \sim P_X$, define the estimator $C_k := \sup_y \frac{1}{k} \sum_i \psi(X_i, y)$. We prove in Theorem A.4 that $\mathbb{E}[C_1] \geq \mathbb{E}[C_2] \geq ... \geq c$, i.e., $C_k$ is in expectation an over-estimator of the sup-partition function $c$. Similarly to the Importance-Weighted ELBO (Burda et al., 2015), the bias of this estimator decreases monotonically as $k \to \infty$, and asymptotically vanishes under regularity assumptions. In light of this, we replace $c$ by $\mathbb{E}[C_k]$ and obtain a $k$-sample under-estimator of the objective $\ell(\theta)$ (which in turn underestimates $F(\lambda)$):

$$\ell_k(\theta) := \mathbb{E}[-\log u_\theta(X)] - \log \mathbb{E}[C_k]; \quad \text{moreover, } \ell_1(\theta) \leq \ell_2(\theta) \leq ... \leq \ell(\theta).$$

In order to apply stochastic gradient ascent, we overcome two more technical hurdles. First, each draw of $C_k$ requires solving a global maximization problem. We note that by restricting to a squared-error $\rho$ and $\mathcal{Y} = \mathcal{X}$, $C_k$ can be computed by finding the mode of a Gaussian mixture density; for this we use the method of Carreira-Perpinan (2000), essentially by hill-climbing from each of the $k$ centroids. Second, to turn $-\log \mathbb{E}[C_k]$ into an expectation, we follow Poole et al. (2019) and underestimate it by linearizing $-\log$ around a scalar parameter $\alpha > 0$, resulting in the following lower bound objective:

$$\tilde{\ell}_k(\theta) := \mathbb{E}[-\log u_\theta(X)] - \mathbb{E}[C_k]/\alpha - \log \alpha + 1. \tag{8}$$

$\tilde{\ell}_k(\theta)$ can finally be estimated by sample averages, and yields a lower bound on the optimal intercept $F(\lambda)$, via $\tilde{\ell}_k(\theta) \leq \ell_k(\theta) \leq \ell(\theta) \leq F(\lambda)$. A trained model $u_{\theta*}$ then yields an R-D lower bound, $R_L(D) = -\lambda D + \tilde{\ell}_k(\theta^*)$. We give a more detailed derivation and pseudocode in Appendix A.4.

## 5 RELATED WORK

**Machine Learning:** The past few years have seen significant progress in applying machine learning to lossy data compression. Theis et al. (2017); Ballé et al. (2017) first showed that a particular type of $\beta$-VAE can be trained to perform data compression using the same objective as Eq. 3. The variational distributions in such a model have shape restrictions to simulate quantization and entropy coding (Ballé et al., 2017). Our upper bound is directly inspired by this line of work, and suggests that such a model can in principle compute the source R-D function when equipped with sufficiently expressive variational distributions and a "rich enough" decoder (see Sec. 3). We note however not all compressive autoencoders admit a probabilistic formulation (Theis et al., 2017); recent work has found training with hard quantization to improve compression performance (Minnen & Singh, 2020), and methods have been developed (Agustsson & Theis, 2020; Yang et al., 2020b) to reduce the gap between approximate quantization at training time and hard quantization at test time. Departing from compressive autoencoders, Yang et al. (2020c) and Flamich et al. (2020) use Gaussian $\beta$-VAEs for data compression and exploit the flexibility of variable-width Gaussian posteriors. Flamich et al. (2020)'s method, and more generally, *reverse channel coding* (Theis & Yosri, 2021), can transmit a sample of $Q_{Z|X}$ with a rate close to that optimized by our upper bound model in Eq. 3, more precisely, $I(X; Z) + \log(I(X; Z) + 1) + O(1)$. i.e., in this one-shot setting (which is standard for

neural image compression), $R(D)$ is no longer achievable; rather, the achievable R-D performance is characterized by $R(D) + \log(R(D) + 1) + O(1)$. Therefore, our R-D bounds can be shifted upwards by this logarithmic factor to give an estimate of the achievable R-D performance in this setting.

Information theory has also broadly influenced unsupervised learning and representation learning. The Information Bottleneck method (Tishby et al., 2000) was directly motivated by, and extends R-D theory and the BA algorithm. Alemi et al. (2018) analyzed the relation between generative modeling and representation learning with a similar R-D Lagrangian to Eq. 2, but used an abstract, model-dependent distortion $\rho(y, x) := -\log p(x|y)$ with an arbitrary $\mathcal{Y}$ and without considering a data compression task. Recently, Huang et al. (2020) proposed to evaluate decoder-based generative models by computing a restricted version of $R_\omega(D)$ (with $Q_Y$ fixed); our result in Sec. 3 ($R_\omega(D) \geq R(D)$) gives a principled way to interpret and compare these model-dependent R-D curves.

**Information Theory:**  While the BA algorithm (Blahut, 1972; Arimoto, 1972) computes the $R(D)$ of a discrete source with a known distribution, no tool currently exists for the general and unknown case. Riegler et al. (2018) share our goal of computing $R(D)$ of a general source, but still require the source to be known analytically and supported on a known reference measure. Harrison & Kontoyiannis (2008) consider the same setup as ours of estimating $R(D)$ of an unknown source from samples, but focus on purely theoretical aspects, assuming prefect optimization. They prove statistical consistency of such estimators for a general class of alphabets and distortion metrics, assuring that our stochastic bounds on $R(D)$ optimized from data samples, when given unlimited computation and samples, can converge to the true $R(D)$. Perhaps closest in spirit to our work is by Gibson (2017), who estimates lower bounds on $R(D)$ of speech and video using Gaussian autoregressive models of the source. However, the correctness of the resulting bounds depends on the modeling assumptions.

A variational lower bound on $R(D)$ was already proposed by Shannon (1959), and later extended (Berger, 1971) to the present version similar to Theorems 4.1 and A.2. In the finite-alphabet case, the maximization characterization of $R(D)$ follows from taking the Lagrange dual of its standard definition in Eq. 1; the dual problem can then be solved by convex optimization (Chiang & Boyd, 2004), but faces the same computational difficulties as the BA algorithm. Csiszár (1974b) proved the general result in Theorem 4.1, applicable to abstract alphabets (in particular, with source $X$ taking values in an arbitrary probability space), by analyzing the fixed-point conditions of the BA algorithm.

## 6 EXPERIMENTS

We estimate the R-D functions of a variety of artificial and real-world data sources, in settings where the BA algorithm is infeasible and no prior known method has been applied. On **Gaussian sources**, our upper bound algorithm is shown to converge to the *exact* R-D function, while our lower bounds become increasingly loose in higher dimensions, an issue we investigate subsequently. We obtain tighter sandwich bounds on **particle physics** and **speech** data than on similar dimensional Gaussians, and compare with the performance of neural compression. We further investigate looseness in the lower bound, experimentally establishing the *intrinsic dimension* of the data as a much more critical contributing factor than the nominal/ambient dimension. Indeed, we obtain tight sandwich bounds on **high-dimension GAN-generated images** with a low intrinsic dimension, and compare with popular neural image compression methods. Finally, we estimate bounds on the R-D function of **natural images**. The intrinsic dimension is likely too high for our lower bound to be useful, while our upper bounds on the Kodak and Tecnick datasets imply at least one dB (in PSNR) of theoretical room for improving state-of-the-art image compression methods, at various bitrates. We also validate our bounds against the BA algorithm over the various data sources, using a 2D marginal of the source to make BA feasible. We provide experimental details and additional results in Appendix A.5 and A.6.

### 6.1 GAUSSIAN SOURCES

We start by applying our algorithms to the factorized Gaussian distribution, one of the few sources with an analytical R-D function. We randomly generate the Gaussian sources in increasing dimensions.

For the upper bound algorithm, we let $Q_Y$ and $Q_{Y|X}$ be factorized Gaussians with learned parameters, predicting the parameters of $Q_{Y|X}$ by a 1-layer MLP encoder. As shown in Fig. 2a-*top*, on a $n = 1000$ dimensional Gaussian (the results are similar across all the $n$ we tested), our upper bound (**yellow**

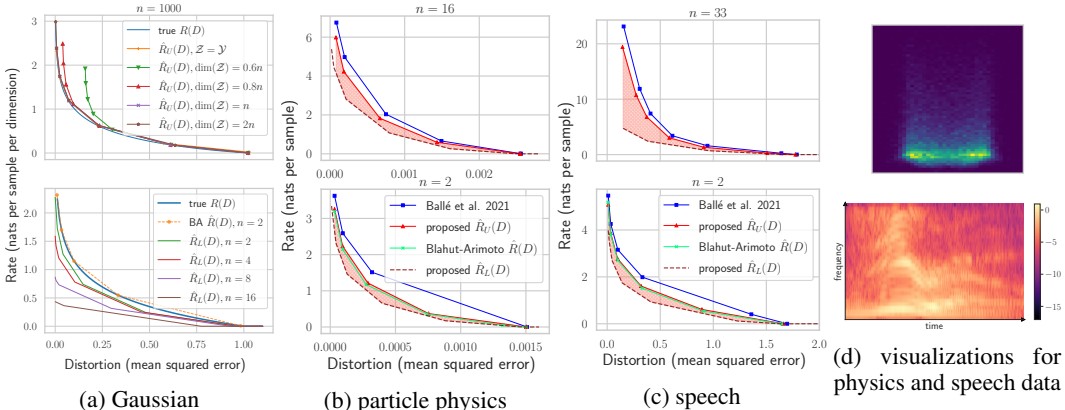

(a) Gaussian  (b) particle physics  (c) speech  (d) visualizations for physics and speech data

Figure 2: **2a** *top*: R-D upper bound estimates on a randomly generated $n =$1000-dimensional Gaussian source; *bottom*: R-D lower bound estimates on standard Gaussians with increasing dimensions (the result of the BA algorithm for $n = 2$ is also shown for reference). **2b** *top*: estimated R-D bounds and the R-D performance of Ballé et al. (2021) on the particle physics dataset; *bottom*: the same experiment but on a 2D marginal distribution of the data, to compare with the BA algorithm. **2c**: the same set of experiments as in **2b**, repeated on the speech dataset. **2d** *top*: histogram of the 2D marginal distribution of the physics data; *bottom*: example spectrogram computed on a speech clip.

curve) accurately recovers the analytical $R(D)$. We also optimized the variational distributions in a latent space $\mathcal{Z}$ with varying dimensions, using an MLP decoder to map from $\mathcal{Z}$ to $\mathcal{Y}$ (see Sec. 3). The resulting bounds are similarly tight when the latent dimension matches or exceeds the data dimension $n$ (**green**, **brown**), but become loose otherwise (**red** and **purple** curves), demonstrating the importance of a rich enough latent space for a tight R-D bound, as suggested by our Theorem A.3.

For the lower bound algorithm, we parameterize $\log u$ by a 2-layer MLP, and study the effect of source dimension $n$ and the number of samples $k$ used in our estimator $C_k$ (and objective $\tilde{\ell}_k$). To simplify comparison of results across different source dimensions, here we consider standard Gaussian sources, whose R-D curve does not vary with $n$ if we scale the rate by $\frac{1}{n}$ (i.e., rate per sample per dimension); the results on randomly generated Gaussians are similar. First, we fix $k = 1024$; Fig. 2a-*bottom* shows that the resulting bounds quickly become loose with increasing source dimension. This is due to the bias of our estimator $C_k$ for the sup-partition function, which causes under-estimation in the objective $\tilde{\ell}_k$. While $C_k$ is defined similarly to an M-estimator (Van der Vaart, 2000), analyzing its convergence behavior is not straightforward, as it depends on the function $u$ being learned. In this experiment, we observe the bias of $C_k$ is amplified by an increase in $n$ or $\lambda$, such that an increasingly large $k$ is required for effective training. In another experiment, we estimate that the $k$ needed to close the gap in the lower bound increases exponentially in $n$; see results in Fig. 4, and a detailed discussion on this, in Appendix A.5.2. Fortunately, as we see in Sec. 6.3, the bias in our lower bound appears to depend on the *intrinsic* rather than (often much higher) nominal dimension of data, giving us a more favorable trade-off between computation and a tighter lower bound as controlled by $k$.

## 6.2 Data from particle physics and speech

The quickly deteriorating lower bound on higher (even 16) dimensional Gaussians may seem disheartening. However, the Gaussian is also the hardest continuous source to compress under squared error (Gibson, 2017), and real-world data often exhibits considerably more structure than Gaussian noise. In this subsection, we experiment on data from particle physics (Howard et al., 2021) and speech (Jackson et al., 2018), and indeed obtain improved sandwich bounds compared to Gaussians.

First, we consider the $Z$-boson decay dataset from Howard et al. (2021), containing $n=$16-dimensional vectors of four-momenta information from independent particle decay events. We ran the neural compression method from (Ballé et al., 2021), as well as our bounding algorithms with similar configurations to before, except we fix $k = 2048$ for the lower bound, and use a normalizing flow for $Q_Z$ in the upper bound model for better expressiveness. Fig. 2b *top* shows the resulting estimated R-D bounds (sandwiched region colored in **red**) and the operational R-D curve for Ballé et al. (2021)

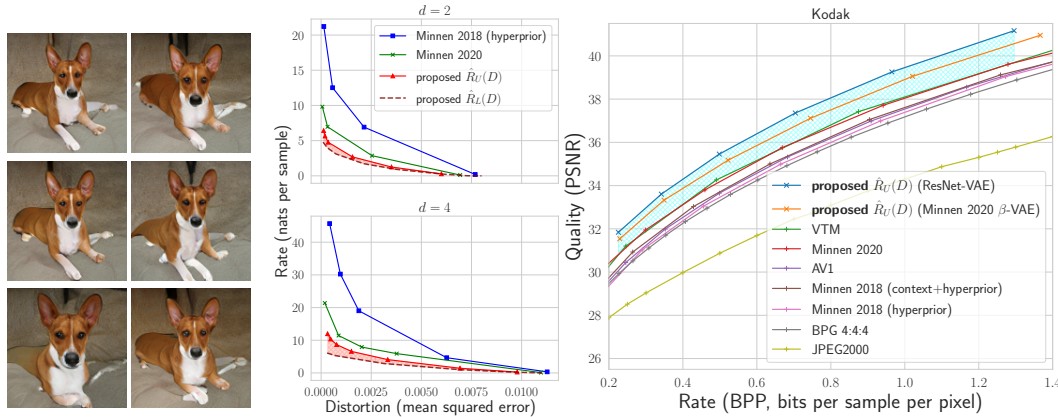

**Figure 3: Left**: 128×128 GAN-generated images, with intrinsic dimension $d = 4$. **Middle**: Bounds on $R(D)$ of GAN images, for $d = 2$ (top) and $d = 4$ (bottom). **Right**: Quality-rate curves of ours and state-of-the-art image compression methods on Kodak (1993), corresponding to R-D upper bounds.

(**blue**). The resulting sandwich bounds appear significantly tighter here than on the Gaussian source with equal dimension ($n = 16$, bottom curve in Fig. 2a *bottom*), and the neural compression method operates with a relatively small average overhead of 0.5 nat/sample relative to our upper bound. To also compare to the ground-truth $R(D)$ as estimated by the BA algorithm, we created a 2-dimensional marginal data distribution (plotted in Fig. 2d *top*), so that the BA algorithm can be feasibly run with a fine discretization grid. As shown in Fig. 2b, the BA estimate of $R(D)$ (**green**) almost overlaps with our upper bound on the 2D marginal, and is tightly sandwiched from below by our lower bound.

We then repeat the same experiments on speech data from the Free Spoken Digit Dataset (Jackson et al., 2018). We constructed our dataset by pre-processing the audio recordings into spectrograms (see, e.g., Fig. 2d *bottom*), then treating the resulting $n = 33$-dimensional frequency feature vectors as independent across time. As shown in Fig. 2c, the gap in our R-D bounds appears wider than on the physics dataset (*top*), but the results on the corresponding 2D marginal appear similar (*bottom*).

## 6.3 THE EFFECT OF INTRINSIC V.S. NOMINAL DIMENSION OF DATA

It is known that learning a manifold has a complexity that depends on the intrinsic dimension of the manifold, but not on its ambient dimension (Narayanan & Mitter, 2010). Our experiments suggest a similar phenomenon for our lower bound, and show that we can still obtain tight sandwich bounds on data with a sufficiently low *intrinsic* dimension despite the high ambient dimension.

First, we explore the effect of increasing the ambient dimension, while keeping the intrinsic dimension of the data fixed. We borrow the 2D banana-source from Ballé et al. (2021), and randomly embed it in $\mathbb{R}^n$. As shown in Fig. 7 and Fig. 8 (in Appendix due to space constraint), our sandwich bounds on the 2D source appear tight, and closely agree with BA (similar to the results in Fig. 2); moreover, the tightness appears unaffected by the increase in ambient dimension $n$ to 4, 16, and 100 (we verified this for $n$ up to 1000). Unlike in the Gaussian experiment, where increasing $n$ required seemingly exponentially larger $k$ for a good lower bound, here a constant $k = 1024$ worked well for all $n$.

Next, we experiment on high-dimension GAN-generated images with varying intrinsic dimension, and obtain R-D sandwich bounds that help assess neural image compression methods. Following Pope et al. (2021), we generate $128 \times 128$ images of `basenji` from a pre-trained GAN, and control the intrinsic dimension by zeroing out all except $d$ dimensions of the noise input to the GAN. As shown in Fig. 3-Left, the images appear highly realistic, showing dogs with subtly different body features and in various poses. We implemented a 6-layer ResNet-VAE (Kingma et al., 2016) for our upper bound model, and a simple convolutional network for our lower bound model. Fig. 3-Middle plots our resulting R-D bounds and the sandwiched region (**red**), along with the operational R-D curves (**blue, green**) of neural image compression methods (Minnen et al., 2018; Minnen & Singh, 2020) trained on the GAN images, for $d = 2$ and $d = 4$. We see that despite the high dimensionality ($n = 128 \times 128 \times 3$), the images require few nats to compress; e.g., for $d = 4$, we estimate $R(D)$ to

be between $\sim 4$ and 8 nats per sample at $D = 0.001$ (30 dB in PSNR). Notably, the R-D curve of Minnen & Singh (2020) stays roughly within a factor of 2 above our estimated true $R(D)$ region. The neural compression methods show improved performance as $d$ decreases from 4 to 2, following the same trend as our R-D bounds. This demonstrates the effectiveness of learned compression methods at adapting to low-dimensional structure in data, in contrast to traditional methods such as JPEG, whose R-D curve on this data does not appear to vary with $d$ and lies orders of magnitude higher.

## 6.4 Natural Images

To establish upper bounds on the R-D function of natural images, we define variational distributions $(Q_Z, Q_{Z|X})$ on a Euclidean latent space for simplicity, and parameterize them as well as a learned decoder $\omega$ via hierarchical VAEs. We consider two VAE architectures: 1. we borrow the convolutional autoencoder architecture of a state-of-the-art image compression model (Minnen & Singh, 2020), but use factorized Gaussians for the variational distributions (we still keep the deep factorized hyperprior, but no longer convolve it with a uniform prior); 2. we also reuse our ResNet-VAE architecture with 6 stochastic layers from the GAN experiments (Sec. 6.3). We trained the models with mean-squared error (MSE) distortion and various $\lambda$ on the COCO 2017 (Lin et al., 2014) images, and evaluated them on the Kodak (1993) and Tecnick (Asuni & Giachetti, 2014) datasets. Following image compression conventions, we report the rate in bits-per-pixel, and the quality (i.e., negative distortion) in PSNR averaged over the images for each $(\lambda, \text{model})$ pair [2]. The resulting *quality-rate* (Q-R) curves can be interpreted as giving upper bounds on the R-D functions of the image-generating distributions. We plot them in Fig. 3, along with the Q-R performance (in actual bitrate) of various traditional and learned image compression methods (Ballé et al., 2017; Minnen et al., 2018; Minnen & Singh, 2020), for the Kodak dataset (see similar results on Tecnick in Appendix Fig. 11). Our $\beta$-VAE version of (Minnen & Singh, 2020) (**orange**) lies on average 0.7 dB higher than the Q-R curves of the original compression model (**red**) and VTM (**green**). With a deeper latent hierarchy, our ResNet-($\beta$-)VAE gives a higher Q-R curve (**blue**) that is on average 1.1 dB above the state-of-the-art Q-R curves (gap shaded in **cyan**). We leave it to future work to investigate which choice of autoencoder architecture and variational distributions are most effective, as well as how the theoretical R-D performance of such a $\beta$-VAE can be realized by a practical compression algorithm (see discussions in Sec. 5).

## 7 Discussions

In this work, we proposed machine learning techniques to computationally bound the rate-distortion function of a data source, a key quantity that characterizes the fundamental performance limit of all lossy compression algorithms, but is largely unknown. Departing from prior work in the information theory community (Gibson, 2017; Riegler et al., 2018), our approach applies broadly to general data sources and requires only i.i.d. samples, making it more suitable for real-world application.

Our upper bound method is a gradient descent version of the classic Blahut-Arimoto algorithm, and closely relates to (and extends) variational autoencoders from neural lossy compression research. Our lower bound method optimizes a dual characterization of the R-D function, which has been known for some time but seen little application outside of theoretical work. Due to difficulties involving global optimization, our lower bound currently requires a squared error distortion for tractability in the continuous case, and is only tight on data sources with a low *intrinsic* dimension. We hope that a better understanding of the lower bound problem will lead to improved algorithms in the future.

To properly interpret bounds on the R-D function, we emphasize that the significance of the R-D function is two-fold: 1. for a given distortion tolerance $D$, no coding procedure can operate with a rate less than $R(D)$, and that 2. this rate is *asymptotically* achievable by a potentially expensive block code. Thus, while a lower bound makes a universal statement about what performance is "too good to be true", the story is more subtle for the upper bound. The achievability proof relies on a random coding procedure that jointly compresses multiple data samples in arbitrarily long blocks (Shannon, 1959). When compressing at a finite block length $b$, $R(D)$ is generally no longer achievable due to a $O(\frac{1}{\sqrt{b}})$ rate overhead (Kontoyiannis, 2000; Kostina & Verdú, 2012). Extending our work to the settings of finite block lengths or non-memoryless sources may be additional future directions.

---

[2]Technically, to estimate an R-D upper bound with MSE as $\rho$, one should compute the distortion by averaging MSE on images; however, the results of many image compression baselines are only available in average PSNR.

ACKNOWLEDGEMENTS

Yibo Yang acknowledges support from the Hasso Plattner Institute at UCI. This material is in part based upon work supported by DARPA under Contract No. HR001120C0021. Stephan Mandt acknowledges support by the National Science Foundation (NSF) under the NSF CAREER Award 2047418; NSF Grants 1928718, 2003237 and 2007719; the Department of Energy under grant DESC0022331, as well as Intel, Disney, and Qualcomm. Any opinions, findings and conclusions or recommendations expressed in this material are those of the author(s) and do not necessarily reflect the views of DARPA or NSF.

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

## A  APPENDIX

### A.1  TECHNICAL DEFINITIONS AND PREREQUISITES

In this work we consider the source to be represented by a random variable $X : \Omega \to \mathcal{X}$, i.e., a measurable function on an underlying probability space $(\Omega, \mathcal{F}, \mathbb{P})$, and $P_X$ is the image measure of $\mathbb{P}$ under $X$. We suppose the source and reproduction spaces are standard Borel spaces, $(\mathcal{X}, \mathcal{A}_\mathcal{X})$ and $(\mathcal{Y}, \mathcal{A}_\mathcal{Y})$, equipped with sigma-algebras $\mathcal{A}_\mathcal{X}$ and $\mathcal{A}_\mathcal{Y}$, respectively. Below we use the definitions of standard quantities from Polyanskiy & Wu (2014).

**Conditional distribution**  The notation $Q_{Y|X}$ denotes an arbitrary conditional distribution (also known as a Markov kernel), i.e., it satisfies

1. For any $x \in \mathcal{X}$, $Q_{Y|X=x}(\cdot)$ is a probability measure on $\mathcal{Y}$;

2. For any measurable set $B \in \mathcal{A}_\mathcal{Y}$, $x \to Q_{Y|X=x}(B)$ is a measurable function on $\mathcal{X}$.

**Induced joint and marginal measures**  Given a source distribution $P_X$, each test channel distribution $Q_{Y|X}$ defines a joint distribution $P_X Q_{Y|X}$ on the product space $\mathcal{X} \times \mathcal{Y}$ (equipped with the usual product sigma algebra, $\mathcal{A}_\mathcal{X} \times \mathcal{A}_\mathcal{Y}$) as follows:

$$P_X Q_{Y|X}(E) := \int_\mathcal{X} P_X(dx) \int_\mathcal{Y} \mathbf{1}\{(x,y) \in E\} Q_{Y|X=x}(dy),$$

for all measurable sets $E \in \mathcal{A}_\mathcal{X} \times \mathcal{A}_\mathcal{Y}$. The induced $y$-marginal distribution $P_Y$ is then defined by

$$P_Y(B) = \int_\mathcal{X} Q_{Y|X=x}(B) P_X(dx),$$

for all measurable sets $\forall B \in \mathcal{A}_\mathcal{Y}$.

**KL Divergence**  We use the general definition of Kullback-Leibler (KL) divergence between two probability measures $P, Q$ defined on a common measurable space:

$$KL(P\|Q) := \begin{cases} \int \log \frac{dP}{dQ} dP, & \text{if } P \ll Q \\ \infty, & \text{otherwise.} \end{cases}$$

$P \ll Q$ denotes that $P$ is absolutely continuous w.r.t. $Q$ (i.e., for all measurable sets $E$, $Q(E) = 0 \implies P(E) = 0$). $\frac{dP}{dQ}$ denotes the Radon-Nikodym derivative of $P$ w.r.t. $Q$; for discrete distributions, we can simply take it to be the ratio of probability mass functions; and for continuous distributions, we can simply take it to be the ratio of probability density functions.

**Mutual Information**  Given $P_X$ and $Q_{Y|X}$, the mutual information $I(X; Y)$ is defined as

$$I(X; Y) := KL(P_X Q_{Y|X} \| P_X \otimes P_Y) = \mathbb{E}_{x \sim P_X}[KL(Q_{Y|X=x} \| P_Y)],$$

where $P_Y$ is the $Y$-marginal of the joint $P_X Q_{Y|X}$, $P_X \otimes P_Y$ denotes the usual product measure, and $KL(\cdot\|\cdot)$ is the KL divergence.

For the mutual information upper bound, it's easy to show that

$$\mathcal{I}_U(Q_{Y|X}, Q_Y) := \mathbb{E}_{x \sim P_X}[KL(Q_{Y|X=x} \| Q_Y)] = I(X; Y) + KL(P_Y \| Q_Y), \qquad (9)$$

so the bound is tight when $P_Y = Q_Y$.

**Obtaining $R(D)$ through the Lagrangian.**  For each $\lambda \geq 0$, we define the Lagrangian by incorporating the distortion constraint in the definition of $R(D)$ through a linear penalty:

$$\mathcal{L}(Q_{Y|X}, \lambda) := I(X; Y) + \lambda \mathbb{E}_{P_X Q_{Y|X}}[\rho(X, Y)], \qquad (10)$$

and define its infimum w.r.t. $Q_{Y|X}$ by the function

$$F(\lambda) := \inf_{Q_{Y|X}} I(X; Y) + \lambda \mathbb{E}[\rho(X, Y)]. \qquad (11)$$

Geometrically, $F(\lambda)$ is the maximum of the $R$-axis intercepts of straight lines of slope $-\lambda$, such that they have no point above the $R(D)$ curve (Csiszár, 1974b).

Define $D_{min} := \inf\{D' : R(D') < \infty\}$. Since $R(D)$ is convex, for each $D > D_{min}$, there exists a $\lambda \geq 0$ such that the line of slope $-\lambda$ through $(D, R(D))$ is tangent to the $R(D)$ curve, i.e.,

$$R(D') + \lambda D' \geq R(D) + \lambda(D) = F(\lambda), \quad \forall D'.$$

When this occurs, we say that $\lambda$ *is associated to* $D$.

Consequently, the $R(D)$ curve is then the envelope of lines with slope $-\lambda$ and $R$-axis intercept $F(\lambda)$. Formally, this can be stated as:

**Lemma A.1.** (Lemma 1.2, Csiszár (1974b); Lemma 9.7, Gray (2011)). *For every distortion tolerance $D > D_{min}$, where $D_{min} := \inf\{D' : R(D') < \infty\}$, it holds that*

$$R(D) = \max_{\lambda \geq 0} F(\lambda) - \lambda D \tag{12}$$

*We can draw the following conclusions:*

1. *For each $D > D_{min}$, the maximum above is attained iff $\lambda$ is associated to $D$.*

2. *For a fixed $\lambda$, if $Q^*_{Y|X}$ achieves the minimum of $\mathcal{L}(\cdot, \lambda)$, then $\lambda$ is associated to the point $(\rho(Q^*_{Y|X}), \mathcal{I}(Q^*_{Y|X}), )$; i.e., there exists a line with slope $-\lambda$ that is tangent to the $R(D)$ curve at $(\rho(Q^*_{Y|X}), \mathcal{I}(Q^*_{Y|X}))$, where we defined the shorthand $\rho(Q_{Y|X}) := \mathbb{E}[\rho(X,Y)]$ and $\mathcal{I}(Q_{Y|X}) := I(X;Y)$ as induced by $P_X Q_{Y|X}$.*

## A.2 FULL VERSION OF THEOREM 4.1

**Theorem A.2.** (Theorem 2.3, Csiszár (1974b); Theorem 1, Kostina (2016).) *Suppose that the following basic assumptions are satisfied.*

1. *$R(D)$ is finite for some $D$, i.e., $D_{min} := \inf\{D : R(D) < \infty\} < \infty$;*

2. *The distortion metric $\rho$ is such that there exists a finite set $E \subset \mathcal{Y}$ such that*

$$\mathbb{E}[\min_{y \in E} \rho(X, y)] < \infty$$

*Then, for each $D > D_{min}$, it holds that*

$$R(D) = \max_{g(x), \lambda} \{\mathbb{E}[-\log g(X)] - \lambda D\} \tag{13}$$

*where the maximization is over $g(x) \geq 0$ and $\lambda \geq 0$ satisfying the constraint*

$$\mathbb{E}\left[\frac{\exp(-\lambda\rho(X,y))}{g(X)}\right] = \int \frac{\exp(-\lambda\rho(x,y))}{g(x)} dP_X(x) \leq 1, \forall y \in \mathcal{Y} \tag{14}$$

Note: the basic assumption 2 is trivially satisfied when the distortion $\rho$ is bounded from above; the maximization over $g(x) \geq 0$ can be restricted to only $1 \geq g(x) \geq 0$. Unless stated otherwise, we use log base $e$ in this work, so the $R(D)$ above is in terms of nats (per sample).

Theorem A.2 can be seen as a consequence of Theorem 4.1 in conjunction with Lemma A.1. We implemented an early version of our lower bound algorithm based on Theorem A.2, generating each R-D lower bound by fixing a target $D$ value and optimizing over both $\lambda$ and $g$ as in Equation 13. However, the algorithm often diverged due to drastically changing $\lambda$. We therefore based our current algorithm on Theorem 4.1, producing R-D lower bounds by fixing $\lambda$ and only optimizing over $g$ (or $u$, in our formulation).

## A.3 THEORETICAL RESULTS

**Theorem A.3.** (A decoder-induced R-D function bounds the source R-D function from above). *Let $X \sim P_X$ be a memoryless source subject to lossy compression under distortion $\rho$. Let $\mathcal{Z}$ be any*

*measurable space ("latent space" in a VAE), and $\omega : \mathcal{Z} \to \mathcal{Y}$ any measurable function ("decoder" in a VAE). This induces a new lossy compression problem with $\mathcal{Z}$ being the reproduction alphabet, under a new distortion function $\rho_\omega : \mathcal{X} \times \mathcal{Z} \to [0, \infty), \rho_\omega(x, z) = \rho(x, \omega(z))$. Define the corresponding $\omega$-dependent rate-distortion function*

$$R_\omega(D) := \inf_{Q_{Z|X} : \mathbb{E}[\rho_\omega(X,Z)] \leq D} I(X;Z) = \inf_{Q_{Z|X} : \mathbb{E}[\rho(X,\omega(Z))] \leq D} I(X;Z).$$

*Then for any $D \geq 0$, $R_\omega(D) \geq R(D)$. Moreover, the inequality is tight if $\omega$ is bijective (so that there is "no information loss").*

*Proof.* Fix $D$. Take any admissible conditional distribution $Q_{Z|X}$ that satisfies $\mathbb{E}[\rho(X, \omega(Z))] \leq D$ in the definition of $R_\omega(D)$. Define a new kernel $Q_{Y|X}$ between $\mathcal{X}$ and $\mathcal{Y}$ by $Q_{Y|X=x} := Q_{Z|X=x} \circ \omega^{-1}, \forall x \in \mathcal{X}$, i.e., $Q_{Y|X=x}$ is the image measure of $Q_{Z|X=x}$ induced by $\omega$. Applying data processing inequality to the Markov chain $X \xrightarrow{Q_{Z|X}} Z \xrightarrow{\omega} Y$, we have $I(X;Z) \geq I(X;Y)$.

Moreover, since $Q_{Y|X}$ is admissible in the definition of $R(D)$, i.e.,

$$\mathbb{E}_{P_X Q_{Y|X}}[\rho(X,Y)] = \mathbb{E}_{P_X Q_{Z|X}}[\rho(X,\omega(Z))] \leq D$$

we therefore have

$$I(X;Z) \geq I(X;Y) \geq R(D) = \inf_{Q_{Y|X} : \mathbb{E}[\rho(X,Y)] \leq D} I(X;Y).$$

Finally, since $I(X;Z) \geq R(D)$ holds for any admissible $Q_{Z|X}$, taking infimum over such $Q_{Z|X}$ gives $R_\omega(D) \geq R(D)$.

To prove $R_\omega(D) = R(D)$ if $\omega$ is bijective, it suffices to show that $R(D) \geq R_\omega(D)$. We use the same argument as before. Take any admissible $Q_{Y|X}$ in the definition of $R(D)$. We can then construct a $Q_{Z|X}$ by the process $X \xrightarrow{Q_{Y|X}} Y \xrightarrow{\omega^{-1}} Z$. Then by DPI we have $I(X;Y) \geq I(X;Z)$. Morever, $Q_{Z|X}$ is admissible: $\mathbb{E}_{P_X Q_{Z|X}}[\rho(X,\omega(Z))] = \mathbb{E}_{P_X Q_{Y|X}}[\rho(X,\omega(\omega^{-1}(Y)))] = \mathbb{E}_{P_X Q_{Y|X}}[\rho(X,Y)] \leq D$. So $I(X;Y) \geq I(X;Z) \geq R_\omega(D)$. Taking infimum over such $Q_{Y|X}$ concludes the proof.

$\square$

*Remark.* Although $\omega^{-1}(\omega(\circ)) = \text{Identity}(\circ)$ for any injective $\omega$, our construction of $Q_{Z|X}$ in the last argument requires the inverse of $\omega$ to exist, so that additionally $\omega(\omega^{-1}(Y)) = Y$. Several learned image compression methods have advocated for the use of sub-pixel convolution, i.e, convolution followed by (invertible) reshaping of the results, over upsampled convolution in the decoder, in order to produce better reconstructions (Theis et al., 2017; Cheng et al., 2020). This can be seen as making the decoder more bijective, therefore reducing the gap of $R_\omega(D)$ over $R(D)$, in light of our above theorem.

**Corollary A.3.1.** (A suitable $\beta$-VAE defines an upper bound on the source R-D function). *Let $X \sim P_X$ be a data source, $\rho : \mathcal{X} \times \mathcal{Y} \to [0, \infty)$ a distortion function, and $\mathcal{Z}$ be any latent space. Consider any $\beta$-VAE consisting of a prior distribution $Q_Z$ on $\mathcal{Z}$, and an encoder which specifies a conditional distribution $Q_{Z|X=x}$ for each $x$. Suppose further that the likelihood (observation) model is chosen to have density $p(x|z) \propto \exp\{-\rho(x, \omega(z)\}$ for some decoder function $\omega : \mathcal{Z} \to \mathcal{Y}$ (a common example being an isotropic Gaussian $p(x|z)$ with mean $\omega(z)$ and fixed variance, specified by a squared error distortion).*

*Then the two terms of the negative ELBO — specifically, the "KL term"*

$$\mathcal{R} := \mathbb{E}_{x \sim P_X}[KL(Q_{Z|X=x} \| Q_Z)],$$

*and the "log-likelihood term" up to a constant,*

$$\mathcal{D} := \mathbb{E}_{P_X Q_{Z|X}}[\rho(X,\omega(Z))] = \mathbb{E}_{P_X Q_{Z|X}}[-\log p(X|Z)] + \text{const},$$

*define a point $(\mathcal{D}, \mathcal{R})$ that lies above the source $R(D)$-curve; i.e., $\mathcal{R} \geq R(\mathcal{D})$.*

*Proof.*

$$\mathcal{R} \geq I_{P_X Q_{Z|X}}(X; Z) \geq \inf_{Q'_{Z|X}: \mathbb{E}_{P_X Q'_{Z|X}}[\rho_\omega(X,Z)] \leq \mathcal{D}} I_{P_X Q'_{Z|X}}(X; Z) =: R_\omega(\mathcal{D}) \geq R(\mathcal{D})$$

The first inequality is the variational upper bound on mutual information (Eq. 9), the second inequality follows from the definition of the $\omega$-induced R-D function (and the fact that $\mathbb{E}_{P_X Q_{Z|X}}[\rho_\omega(X, Z)] =:$ $\mathcal{D}$), and the last inequality is Theorem A.3.

$\square$

*Remark.* As also remarked by Theis et al. (2017), not all distortion functions lead to the $\beta$-VAE interpretation above. For this to work, the function $\exp\{-\rho(x, \omega(z))\}$ must be normalizable in $x$ (w.r.t. a suitable base measure on $\mathcal{X}$) for every $z$, and the normalizing constant may not depend on $z$.

**Theorem A.4.** (Basic properties of the proposed estimator $C_k$ of the sup-partition function.) *Let $X_1, X_2, ... \sim P_X$ be a sequence of i.i.d. random variables. Let $\psi : \mathcal{X} \times \mathcal{Y} \to \mathbb{R}^+$ be a measurable function. For each $k$, define the random variable $C_k := \sup_y \frac{1}{k} \sum_i \psi(X_i, y)$. Then*

1. *$C_k$ is an overestimator of the sup-partition function $c$, i.e.,*
   *$\mathbb{E}[C_k] = \mathbb{E}_{X_1,...,X_k}[\sup_y \frac{1}{k} \sum_i \psi(X_i, y)] \geq \sup_y \mathbb{E}[\psi(X, y)] =: c;$*

2. *The bias of $C_k$ decreases with increasing $k$, i.e.,*
   *$\mathbb{E}[C_1] \geq \mathbb{E}[C_2] \geq ... \geq \mathbb{E}[C_k] \geq \mathbb{E}[C_{k+1}] \geq ... \sup_y \mathbb{E}[\psi(X, y)] = c;$*

3. *If $\psi(x, y)$ is bounded and continuous in $y$, and if $\mathcal{Y}$ is compact, then $C_k$ is strongly consistent, i.e., $C_k$ converges to $c$ almost surely (as well as in probability, i.e., $\lim_{k \to \infty} \mathbb{P}(|C_k - c| > \epsilon) = 0, \forall \epsilon > 0$), and $\lim_{k \to \infty} \mathbb{E}[C_k] = c$.*

*Proof.* We prove each in turn:

1. $\mathbb{E}[C_k] = \mathbb{E}[\sup_y \frac{1}{k} \sum_i \psi(X_i, y)] \geq \sup_y \mathbb{E}[\frac{1}{k} \sum_i \psi(X_i, y)] = \sup_y \mathbb{E}[\psi(X, y)] = c$

2. First, note that $\mathbb{E}[C_1] \geq \mathbb{E}[C_k]$ since

$$\mathbb{E}[C_1] = \mathbb{E}[\sup_y \psi(X_1, y)] = \mathbb{E}[\frac{1}{k} \sum_i \sup_y \psi(X_i, y)] \geq \mathbb{E}[\sup_y \frac{1}{k} \sum_i \psi(X_i, y)] = \mathbb{E}[C_k]$$

We therefore have

$$\mathbb{E}[C_{k+1}] = \mathbb{E}[\sup_y \frac{1}{k+1} \sum_{i=1}^{k+1} \psi(X_i, y)]$$

$$= \mathbb{E}[\sup_y \{\frac{1}{k+1} \sum_{i=1}^{k} \psi(X_i, y) + \frac{1}{k+1} \psi(X_{k+1}, y)\}]$$

$$\leq \mathbb{E}[\sup_y \{\frac{1}{k+1} \sum_{i=1}^{k} \psi(X_i, y)\} + \sup_y \{\frac{1}{k+1} \psi(X_{k+1}, y)\}]$$

$$= \frac{k}{k+1} \mathbb{E}[C_k] + \frac{1}{k+1} \mathbb{E}[C_1]$$

$$\leq \mathbb{E}[C_k]$$

3. Define the shorthand $M_k(y) := \frac{1}{k} \sum_{i=1}^{k} \psi(X_i, y)$, $\Psi(y) = \mathbb{E}[\psi(X, y)]$. Denote the sup norm on the set of continuous bounded functions on $\mathcal{Y}$ by $\|f(\cdot)\|_\infty := \sup_{y \in \mathcal{Y}} |f(y)|$. Then it holds that

$$|\sup_y M_k(y) - \sup_y \Psi(y)| = |\sup_y |M_k(y)| - \sup_y |\Psi(y)||$$

$$= |\|M_k(\cdot)\|_\infty - \|\Psi(\cdot)\|_\infty|$$

$$\leq \|M_k(\cdot) - \Psi(\cdot)\|_\infty$$

$$= \sup_y |M_k(y) - \Psi(y)|,$$

where we made use of the fact that $\psi \geq 0$ in the first equality, and the reverse triangle inequality in the second-to-last step. By the uniform strong law of large numbers (Ferguson, 2017),

$$\lim_{k \to \infty} \sup_y |M_k(y) - \Psi(y)| = 0$$

almost surely. Therefore, by the inequality we just showed, we also have

$$\lim_{k \to \infty} |\sup_y M_k(y) - \sup_y \Psi(y)| \leq \lim_{k \to \infty} \sup_y |M_k(y) - \Psi(y)| = 0$$

almost surely, i.e.,

$$\lim_{k \to \infty} C_k = \lim_{k \to \infty} \sup_y M_k(y) = \sup_y \mathbb{E}[\psi(X, y)] := c$$

almost surely. Then it trivially follows that $C_k$ also converges to $c$ in probability, and that $\lim_{k \to \infty} \mathbb{E}[C_k] = c$ by the bounded convergence theorem.

$\square$

## A.4 DETAILED LOWER BOUND METHOD

**Approximations for stochastic gradient ascent.** Recall the lower bound objective obtained by replacing the sup-partition function $c$ by the mean of the $k$-sample underestimator $C_k$:

$$\ell_k(\theta) := \mathbb{E}[-\log u_\theta(X)] - \log \mathbb{E}[C_k]$$

Unfortunately, we still cannot apply stochastic gradient ascent to $\ell_k$, as two more difficulties remain, which we solve by further approximations. First, $C_k$ is still hard to compute, as it is defined through a global maximization problem. We note that by restricting to a symmetric $\rho$ and $\mathcal{Y} = \mathcal{X}$, the maximization objective of $C_k$ has the form of a reweighted kernel density estimate,

$$\frac{1}{k} \sum_{i=1}^{k} \psi(x_i, y) = \frac{1}{k} \sum_{i=1}^{k} \frac{1}{u(x_i)} \exp(-\lambda \rho(x_i, y)),$$

where each data point is reweighted by $\frac{1}{u(x)}$. For a squared error distortion, this becomes a Gaussian mixture density, $\frac{1}{k} \sum_i \psi(x_i, y) \propto \sum_i \pi_i \exp(-\lambda \|x_i - y\|^2)$, with centroids defined by the samples $x_1, ..., x_k$, and mixture weights $\pi_i = (k u(x_i))^{-1}$. The global mode of a Gaussian mixture can generally be found by hill-climbing from each of the $k$ centroids, except in rare artificial examples (Carreira-Perpinan, 2000; 2020); we therefore use this procedure to compute $C_k$, but note that other methods exist (Lee et al., 2019; Carreira-Perpinan, 2007).

The second difficulty is that even if we could estimate $\mathbb{E}[C_k]$ (with samples of $C_k$ computed by global optimization), the objective $\ell_k$ requires an estimate of its logarithm; a naive application of Jensen's inequality $-\log \mathbb{E}[C_k] \geq \mathbb{E}[-\log C_k]$ results in an *over*-estimator (as does the IWAE estimator), whereas we require a lower bound. Following Poole et al. (2019), we use the following basic lower bound of $-\log(x)$ by its linearization around some point $\alpha$, i.e.,

$$-\log(x) \geq -x/\alpha - \log(\alpha) + 1, \forall x, \alpha > 0$$

with equality achieved by $\alpha = x$. This results in the final lower bound objective we optimize:

$$\tilde{\ell}_k(\theta) := \mathbb{E}[-\log u_\theta(X)] - \mathbb{E}[C_k]/\alpha - \log \alpha + 1. \tag{15}$$

Since the linear underestimator of $-\log \mathbb{E}[C_k]$ is tightest near $\alpha = \mathbb{E}[C_k]$, in our algorithm we set $\alpha$ adaptively to an estimate of $\mathbb{E}[C_k]$ (separately from the $\mathbb{E}[C_k]$ term we estimate in the objective function).

**The algorithm.** We give a pseudo-code implementation of the algorithm in Algorithm 1. The code largely follows Python semantics, and assumes an auto-differentiation package is available, such as GradientTape as provided in `Tensorflow`. We use $\gamma_k$ to denote the function in the supremem definition of $C_k$, i.e., $\gamma_k(y) := \frac{1}{k} \sum_{i=1}^{k} \exp\{-\lambda \rho(x_i, y)\} / u_\theta(x_i)$.

**Algorithm 1:** Example implementation of the proposed algorithm for estimating rate-distortion lower bound $R_L(D)$.

---

**1** **Requires:** $\lambda > 0$, model $u_\theta$ (e.g., a neural network) parameterized by $\theta$, batch sizes $k, m$, and gradient ascent step size $\epsilon$.

**2** **while** $\tilde{\ell}_k$ *not converged* **do**

    `// Draw 2m samples of` $C_k$`; use the last m samples to set` $\alpha$`,`
        `and the first m samples to estimate` $\mathbb{E}[C_k]$ `in` $\tilde{\ell}_k$`.`

**3**     **for** $j := 1$ **to** $2m$ **do**

**4**         Draw $k$ data samples, $\{x_1^j, ..., x_k^j\}$

**5**         $y^j, C_k^j := \text{compute\_}C_k(\theta, \lambda, \{x_1, ..., x_k\})$

**6**     **end**

**7**     $\alpha := \frac{1}{m} \sum_{j=m+1}^{2m} C_k^j$

    `// Compute objective` $\tilde{\ell}_k(\theta)$ `and update` $\theta$ `by gradient ascent.`

**8**     with GradientTape() as tape:

**9**         $E := \frac{1}{m} \sum_{j=1}^{m} \gamma_k(y^j, \theta, \{x_1^j, ..., x_k^j\})$ `// Estimate of` $\mathbb{E}[C_k]$

**10**         $\tilde{\ell}_k := -\frac{1}{m} \sum_{j=1}^{m} \frac{1}{k} \sum_{i=1}^{k} \log u_\theta(x_i^j) - \frac{1}{\alpha} E - \log \alpha + 1$

**11**         gradient := tape.gradient($\tilde{\ell}_k, \theta$)

**12**     $\theta := \theta + \epsilon$ gradient

**13** **end**

**14** **Subroutine** $\gamma_k(y, \theta, \lambda, \{x_1, ..., x_k\})$ :

    `// Evaluate the global maximization objective at` $y$`.`

**15**     Return $\frac{1}{k} \sum_{i=1}^{k} \exp\{-\lambda\rho(x_i, y)\}/u_\theta(x_i)$

**16** **Subroutine** compute\_$C_k(\theta, \lambda, \{x_1, ..., x_k\})$ :

    `// Compute the global optimum of` $\gamma_k(y)$`, assuming squared`
        `distortion` $\rho(x, y) \propto \|x - y\|^2$`.`

**17**     opt\_loss := $-\infty$

**18**     **for** $i := 1$ **to** $k$ **do**

        `// Run gradient ascent from the` $i$`th mixture component.`

**19**         $y := x_i$

**20**         **while** $y$ *not converged* **do**

**21**             with GradientTape() as tape:

**22**                 loss := $\gamma_k(y, \theta, \lambda, \{x_1, ..., x_k\})$

**23**                 gradient := tape.gradient(loss, $y$)

**24**                 $y := y + \epsilon$ gradient

**25**         **end**

**26**         **if** *loss > opt\_loss* **then**

**27**             opt\_loss := loss

**28**             $y^* := y$

**29**         **end**

**30**     **end**

**31**     $C_k = $ opt\_loss

**32**     return $(y^*, C_k)$

---

For a given $\lambda > 0$ and model $u_\theta : x \to \mathbb{R}^+$, the lower bound algorithm works by (stochastic) gradient ascent on the objective $\tilde{\ell}_k(\theta)$ (Eq. 8) w.r.t. the model parameters $\theta$. To compute the gradient, we need an estimate of $\mathbb{E}[C_k]$ (ultimately $\log \mathbb{E}[C_k]$) as part of the objective, which requires us to draw $m$ samples of $C_k$. In the pseudo-code, a separate set of $m$ samples of $C_k$ are also drawn to set the linear expansion point $\alpha$. In our actual implementation, we set $\alpha$ to an exponential moving average of $\mathbb{E}[C_k]$ estimated from previous iterations (e.g., replacing line 7 of the pseudo-code by $\alpha := 0.2\alpha + 0.8E$), so only $m$ samples of $C_k$ need to be drawn for each iteration of the algorithm. We did not find this approximation to significantly affect the training or results.

Recall $C_k(\theta)$ is defined by the result of maximizing w.r.t. $y$. When computing the gradient with respect to $\theta$, we differentiate through the maximization operation by evaluating $C_k = \gamma_k(y^*)$ on

the forward pass, using the argmax $y^*$ found by the subroutine compute_$C_k$. This is justified by appealing to a standard envelope theorem (Milgrom & Segal, 2002).

By Lemma A.1, each $u_\theta^*$ trained with a given value of $\lambda$ yields a linear under-estimator of $R(D)$,

$$R_L^\lambda(D) = -\lambda D + \tilde{\ell}_k(\theta^*).$$

We obtain the final $R_L(D)$ lower bound by taking the upper envelope of all such lines, i.e.,

$$R_L(D) := \max_{\lambda \in \Lambda} R_L^\lambda(D),$$

where $\Lambda$ is the set of $\lambda$ values we trained with.

Tips and tricks:

To avoid numerical issues, we always parameterize $\log u$, and all the operations involving $C_k$ and $\alpha$ are performed in the log domain; e.g., we optimize $\log \gamma_k$ instead of $\gamma_k$ (which does not affect the result since $\log$ is monotonically increasing), and use logsumexp when taking the average of multiple $C_k$ samples in the log domain.

During training, we only run an approximate version of the global optimization subroutine compute_$C_k$ for efficiency, as follows. Instead of hill-climbing from each of the $k$ mixture component centroids, we only do so from the $t$ highest-valued centroids under the objective $\gamma_k$, for a small number of $t$ (say 10). This approximation is not used when reporting the final results.

### A.5 ADDITIONAL EXPERIMENTAL DETAILS AND RESULTS

#### A.5.1 COMPUTATIONAL ASPECTS

Our methods are implemented using the `Tensorflow` library. Our experiments with learned image compression methods additionally used the `tensorflow-compression`[3] library. Our experiments on images were run on Titan RTX GPUs, while the rest of the experiments were run on Intel(R) Xeon(R) CPUs. We used the Adam optimizer for gradient based optimization in all our experiments, typically setting the learning rate between $1e - 4$ and $5e - 4$. Training the $\beta$-VAEs for the upper bounds required from a few thousand gradient steps on the lower-dimension problems (under an hour), to a few million gradient steps on the image compression problem (a couple of days; similar to reported in Minnen & Singh (2020)). With our approximate mode finding procedure (starting hill climbing only from a small number of centroids, see Sec. A.4), the lower bound models required less than 10000 gradient steps to converge in all our experiments.

#### A.5.2 GAUSSIANS

**Data** Here the data source is an $n$-dimensional Gaussian distribution with a diagonal covariance matrix, whose parameters are generated randomly. We sample each dimension of the mean uniformly randomly from $[-0.5, 0.5]$ and variance from $[0, 2]$. The ground-truth R-D function of the source is computed analytically by the reverse water-filling algorithm (Cover & Thomas, 2006).

**Upper Bound Experiments** For the $\mathcal{Z} = \mathcal{Y}$ (no decoder) experiment, we chose $Q_Y$ and $Q_{Y|X}$ to be factorized Gaussians; we let the mean and variance vectors of $Q_Y$ be trainable parameters, and predict the mean and variance of $Q_{Y|X=x}$ by one fully connected layer with $2n$ output units, using softplus activation for the $n$ variance components.

For our experiments involving a decoder, we parameterize the variational distributions $Q_Z$ and $Q_{Z|X}$ similarly to before, and use an MLP decoder with one hidden layer (with the number of units equal the data dimension $n$, and leaky ReLU activation) to map from the $\mathcal{Z}$ to $\mathcal{Y}$ space. We observe the best performance with a linear (or identity) decoder and a simple linear encoder; using deeper networks with nonlinear activation required more training iterations for SGD to converge, and often to a poorer upper bound. In fact, for a factorized Gaussian source, it can be shown analytically that the optional $Q_{Y|X=x}^*$ is a Gaussian whose mean depends linearly on $x$, and an identity (no) decoder is optimal.

---

[3] https://github.com/tensorflow/compression

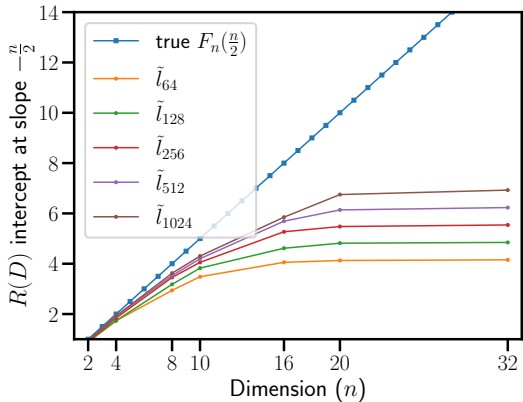

Figure 4: $R$-axis intercept estimates at (negative) slope $\lambda = \frac{n}{2}$ from our lower bound algorithm, trained with increasing $n$ and $k$. The $R$-axis intercept of an optimally tight linear lower bound, $F_n(\frac{n}{2})$, increases linearly as a function of $n$; however, we see that the best achievable $R$-axis intercept achieved with a particular setting of $k$, $\tilde{\ell}_k$, plateaus out at $\log_k$ as $n$ increases.

**Lower Bound Experiments**  In our lower bound experiments, we parameterize $\log u$ by an MLP with 2 hidden layers with $20n$ hidden units each and SeLU activation, where $n$ is the dimension of the Gaussian source. We fixed $k = 1024$ for the results with varying $n$ in Figure 2a-*bottom*. We vary both $k$ and $n$ in the additional experiment below.

To investigate the necessary $k$ needed to achieve a tight lower bound, and its relation to $\lambda$ or the Gaussian source dimension $n$, we ran a series of experiments with $k$ ranging from 64 to 1024 on increasingly high dimensional standard Gaussian sources, each time setting $\lambda = \frac{n}{2}$. For an $n$-dimensional standard Gaussian, the true $R$-intercept, $F_n(\lambda)$, has an analytical form; in particular, $F_n(\frac{n}{2}) = \frac{n}{2}$. In Figure 4, we plot the final objective estimate, $\tilde{\ell}_k$, using the converged MLP model, one for each $k$ and $n$. As we can see, the maximum achievable $\tilde{\ell}_k$ plateaus to the value $\log k$ as we increase $n$, and for sufficiently high dimension (e.g., $n = 20$ here), doubling $k$ only brings out a constant ($\log 2$) improvement in the objective. This phenomenon relates to the over-estimation bias of $C_k$ when $k$ is too low compared to $\lambda$ or the "spread" of the data, and can be understood as follows.

For a given sample size $k$, there is a "catastrophic" regime produced by increasingly large $\lambda$ (corresponding to an exceedingly narrow Gaussian kernel), or increasingly high (intrinsic) dimension of the data, so that the $k$ data samples appear very far apart. Numerically, this is exhibited by very quick termination of the $k$ hill-climbing runs when computing $C_k$ (subroutine compute_$C_k$ in Algorithm. 1), since the mixture components are well separated, and the mixture centroids are nearly stationary points of the mixture density. The value of the mixture density at each centroid $x_i$ can be approximated as $\gamma_k(x_i) = \frac{1}{k}\frac{1}{u(x_i)} + \frac{1}{k}\sum_{j\neq i}\exp\{-\lambda\rho(x_j,x_i)\}/u_\theta(x_j) \approx \frac{1}{ku(x_i)}$ , since $\exp\{-\lambda\rho(x_j,x_i)\} \approx 0$ for all $j \neq i$. The maximization problem defining $C_k$ then essentially reduces to checking which mixture component is the highest, and returning the corresponding centroid (or a point very close to it), so that $C_k \approx \sup_{i=1,\ldots,k}\frac{1}{ku(x_i)}$. This implies

$$\mathbb{E}[C_k] \approx \mathbb{E}[\sup_{i=1,\ldots,k}\frac{1}{ku(X_i)}] \leq \mathbb{E}[\sup_x \frac{1}{ku(x)}] = \sup_x \frac{1}{ku(x)},$$

therefore

$$\ell_k := \mathbb{E}[-\log u(X)] - \log\mathbb{E}[C_k] \tag{16}$$

$$\leq \mathbb{E}[-\log u(X)] + \log k + \inf_x \log u(x) \tag{17}$$

$$= \log k + \mathbb{E}[-\log u(X) + \inf_{x'} \log u(x')] \leq \log k. \tag{18}$$

Thus, when these approximations hold, the maximum achievable lower bound objective $\tilde{\ell}_k$ cannot exceed $\log k$. On sources such as high-dimension (e.g., $n = 10000$) Gaussians, or $256 \times 256$ patches of natural images, $\lambda$ needs to be on the order of $10^6$ or higher to target even the low-distortion region

of the R-D curve, and the above analysis well describes the behavior of the lower bound algorithm for any value of $k$ we feasibly experimented with.

### A.5.3 PARTICLE PHYSICS

**Data** We borrowed the $Z \to e^+e^-$ dataset from Howard et al. (2021), containing 331699 data observations. Each $n = 16$-dimensional data vector corresponds to an independent $Z$-boson decay event, and is the concatenation of the four-momenta of an electron ($e^-$) and positron ($e^+$) both in the theoretical prior space and observed space. The 2D marginal distribution (where we additionally compare with the BA algorithm) is formed by taking the two data coordinates with the least absolute value of correlation. A histogram is given in Figure 5a.

Unlike in the Gaussian experiments, where we generate new random samples from the source on the fly for training and evaluation, here the true underlying source is only approximated by a finite number of samples in the given dataset. Since our R-D upper/lower bounds have the form of expectations of learned functions w.r.t. the *true* underlying source (explained in more detail in Section. A.6), we create a separate held-out test set of 10000 samples for our final estimates of R-D bounds using the trained models. We also use a small subset of the training data as a validation set for model selection, typically using the most expressive model we can afford without overfitting. We use the same train/test split and the same model architectures on the 2D marginal data as on the $n = 16$ data.

**Upper Bound Experiments** For our R-D upper bound model, we use a VAE with a MLP decoder, a normalizing flow "prior" $Q_Z$, and an MLP encoder computing a factorized Gaussian $Q_{Z|X}$. We set the latent space to have the same dimensionality $n$ as the data. The encoder and decoder MLPs have three layers with $[500, 500, l]$ units in each, where $l = n$ for the deocder MLP and $l = 2n$ for the encoder MLP (as it computes both a mean and a variance vector for $Q_{Z|X}$); we use softplus activation in each hidden layer. $Q_Z$ is parameterized as a base Gaussian distribution transformed by three stacks of MAF (Papamakarios et al., 2017).

For the Nonlinear Transform Coding (NTC) (Ballé et al., 2021) model, we use a VAE with a similar 500-500-$n$ MLP architecture for the encoder and decoder (thus the latent space has the same dimensionality $n$ as the data), but modify the variational distributions to simulate quantization and end-to-end compression. As in (Ballé et al., 2018; Ballé et al., 2021), we let $Q_{Z|X}$ be a factorized uniform distribution with unit width, and $Q_Z$ be a factorized neural density model convolved with a uniform noise.

For the BA algorithm on the 2D marginal, we obtain discretized alphabets and source probabilities as follows. First, we determine the smallest box set $S$ in $\mathbb{R}^2$ containing all the data samples (simply the Cartesian product of sample ranges along the x and y axes); next, we finely tile up $S$ using small rectangle bins, and compute a histogram of data samples over them. We then let the source and reproduction alphabets to be the bin centers, and set the source distribution to be the frequency of samples in each bin. Finally, for computation efficiency, we remove the bin centers from the source alphabet associated with 0 samples; this does not affect the results.

We ran a maximum of 2000 steps of the BA algorithm on the test set, terminating early if the objective has not improved by more than 0.0001% in consecutive steps.

**Lower Bound Experiments** We parameterize our lower bound model $\log u$ by an MLP with 3 hidden layers with 200 hidden units each and SeLU activation. We fixed $k = 2048$.

### A.5.4 SPEECH

**Data** We use the Free Spoken Digit Dataset (Jackson et al., 2018) consisting of recordings of spoken digits from 'zero' to 'nine' at a sampling rate of 8kHz. We use a subset of the recordings with a randomly chosen speaker id ('theo'), and keep the original train-test split [4] consisting of 450 and 50 audio files for the train and test sets, respectively. From each audio file, we extracted Short-time Fourier Transform (STFT) coefficients using Tensorflow's `tf.signal.stft` routine with a 63-sample FFT window (`frame_length=63`) in 1-sample steps (`frame_step=1`), took the magnitude and converted to log domain, a common pre-processing step for computing spectrograms

---

[4]`https://github.com/Jakobovski/free-spoken-digit-dataset/#usage`

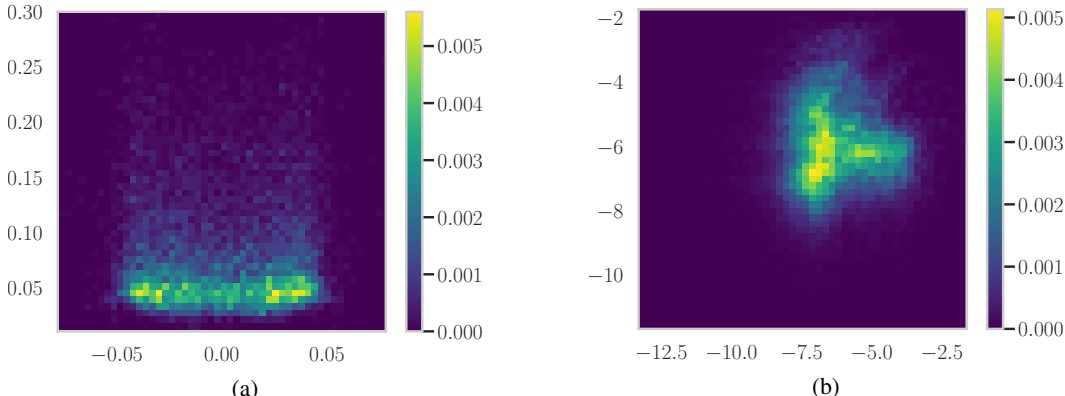

Figure 5: Normalized histograms of the 2D marginals of the particle physics (5a) and speech (5b) datasets. The quantization bins and the associated empirical PMFs are also used define the discretized 2D source distributions on which the BA algorithm is run.

and Mel-frequency cepstral coefficients. The resulting $n = 33$-dimensional Fourier feature vectors are then shuffled across time steps and treated as i.i.d. samples in our experiments, with over 1 million samples in the training set and 120000 samples in the test set. We follow the same procedure as in the physics experiment, creating a 2D marginal from the two coordinates with the least absolute correlation; a histogram is given in Figure 5b.

**Experiments** We use the same model architectures and experimental procedures/hypermarameters as in the physics experiment.

### A.5.5  BANANA-SHAPED SOURCE

**Data** The 2D banana-shaped source (Ballé et al., 2021) is a RealNVP transform of a 2D Gaussian [5]. See Figure 6 for a plot of its density function.

We embed the banana source in a higher dimension $n$ by multiplying with a randomly sampled $n \times 2$ matrix. This is done by simply passing samples of the 2D banana source through a linear MLP layer with $n$ output units and no activation, with its weight matrix randomly drawn from a Gaussian by the Glorot initialization procedure.

**Upper Bound Experiments** We reproduced the NTC experiment using the same compressive autoencoder as in Ballé et al. (2021), with a 2-dimensional latent space and two-layer MLPs for both the encoder and decoder, with 100 hidden units and softplus activation in each hidden layer. For our upper bound model, we use the same MLP architecture as in Ballé et al. (2021), but parameterize $Q_Z$ by a MAF (Papamakarios et al., 2017) and $Q_{Z|X}$ by a factorized Gaussian (doubling the number of output of the encoder), similar to on previous experiments.

For the higher-dimension embeddings of the banana-shaped source, we use the same $\beta$-VAE as on the 2D source, but set the number of hidden units in each hidden layer to $100n$, where $n$ is the dimension of the source. We cap the number of hidden units per layer at 2000 for $n > 20$, and do not find this to adversely affect the results.

**Lower Bound Experiments** We use an MLP with three hidden layers and SeLU activations for the $\log u$ model; as in the upper bound models, here we set the number of hidden units in each layer to be $100n$, and cap it at 1000. In all the experiments we set $k = 1024$.

---

[5]Source code taken from the authors' GitHub repo https://github.com/tensorflow/compression/blob/66228f0faf9f500ffba9a99d5f3ad97689595ef8/models/toy_sources/toy_sources.ipynb.

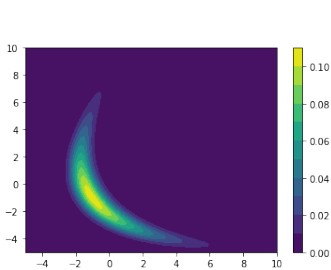

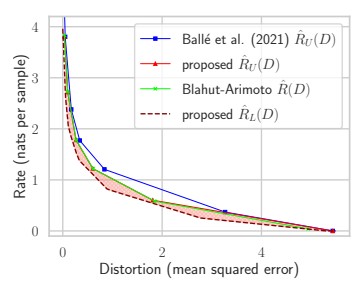

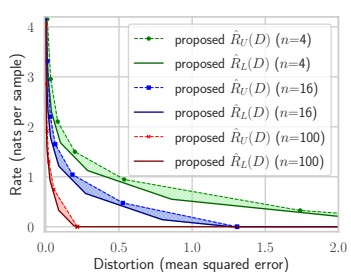

Figure 6: Density plot of the 2D banana-shaped distribution from Ballé et al. (2021).

Figure 7: R-D sandwich bounds and the R-D performance of non-linear transform coding (Ballé et al., 2021) on the banana-shaped source. For reference, the BA algorithm is also run on a fine discretization of the source.

Figure 8: R-D sandwich bounds on higher-dimension embeddings of the banana-shaped source. The tightness of our bounds appear unaffected by the increasing $n$.

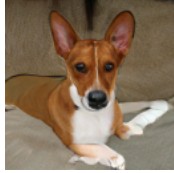 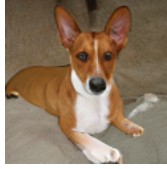 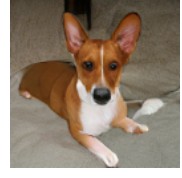 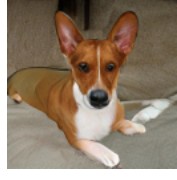 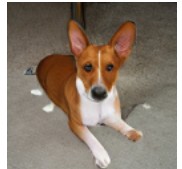

Figure 9: Random samples of basenji from a BigGAN, $d = 2$.

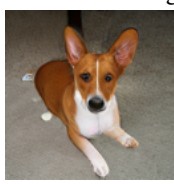 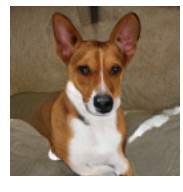 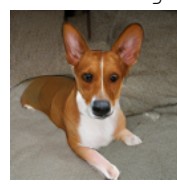 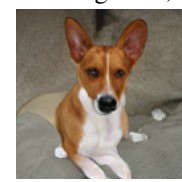 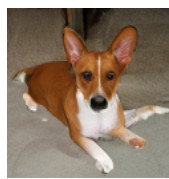

Figure 10: Random samples of basenji from a BigGAN, $d = 4$.

As illustrated by Figure 8, the tightness of our sandwich bounds do not appear to be affected by the dimension $n$ of the data (we verified this up to $n = 1000$), unlike on the Gaussian experiment where for a fixed $k$ the lower bound became increasingly loose with higher $n$.

### A.5.6 GAN-GENERATED IMAGES

**Data** We adopt the same setup as in the GAN experiment by Pope et al. (2021). We use a BigGAN (Brock et al., 2019) pretrained on ImageNet at $128 \times 128$ resolution (so that $n = 128 \times 128 \times 3 = 49152$), downloaded from https://tfhub.dev/deepmind/biggan-deep-128/1. To generate an image of an ImageNet category, we provide the corresponding one-hot class vector as well as a noise vector to the GAN. Following Poole et al. (2019), we control the intrinsic dimension $d$ of the generated images by setting all except the first $d$ dimensions of the 128-dimension noise vector to zero, and use a truncation level of $0.8$ for the sample diversity factor. Since the GAN generates values in $[-1, 1]$, we rescale them linearly to $[0, 1]$ to correspond to images, so that the alphabets $\mathcal{X} = \mathcal{Y} = [0, 1]^n$. As in (Pope et al., 2021), we experimented with images from the ImageNet class basenji. In Figure 9 and 10, we plot additional random samples, for $d = 2$ and $d = 4$.

**Upper Bound Experiments** We implemented a version of ResNet-VAE following the appendix of Kingma et al. (2016), using the $3 \times 3$ convolutions of Cheng et al. (2020) in the encoder and decoder of our model. Our model consists of 6 layers of latent variables, and implements bidirectional inference using factorized Gaussian variational distributions at each stochastic layer. During an inference pass, an input image goes through 6 stages of convolution followed by downsampling (each time reducing the height and width by 2), and results in a stochastic tensor at each stage. In encoding

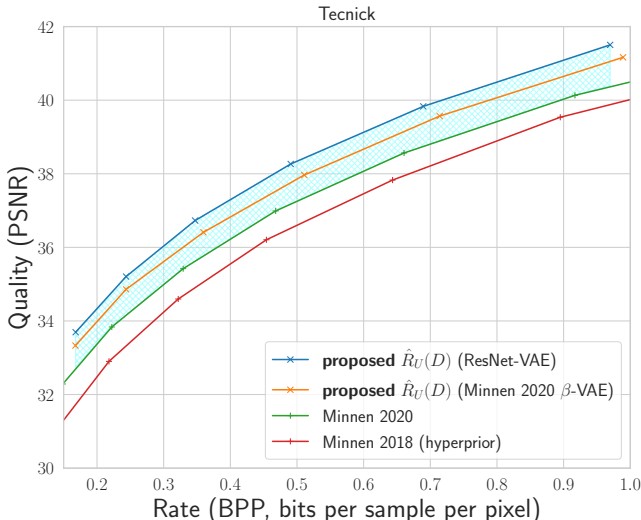

Figure 11: Quality-rate curves on the Tecnick dataset.

order, the stochastic tensors have decreasing spatial extent and an increasing number of channels equal to 4, 8, 16, 32, 64, and 128. The latent tensor $Z_0$ at the topmost generative hierarchy is flattened and modeled by a MAF prior $Q_{Z_0}$ (aided by the fact that all the images have a fixed shape).

**Lower Bound Experiments** We parameterize the $\log u$ model by a simple feedforward convolutional neural network. It contains three convolutional layers with 4, 8, and 16 filters, each time downsampling by 2, followed by a fully connected layer with 1024 hidden units. Again we use SeLU activation in the hidden units, and set $k = 2048$ in all the experiments.

### A.5.7 NATURAL IMAGES

**Data** For signals such as images or video, which can have arbitrarily high spatial dimension, it is not immediately clear how to define i.i.d. samples. But since our focus is on image compression, we follow the common practice of training convolutional autoencoders on random 256×256-pixel crops of high resolution images as in learned image compression research (Ballé et al., 2017), noting that current methods cannot effectively exploit spatial redundancies larger than this scale (Minnen & Singh, 2020). We can then use the trained models to estimate R-D upper bounds on the image-generating distributions underlying standard benchmark datasets such as Kodak Kodak (1993) and Tecnick Asuni & Giachetti (2014).

As a representative dataset of natural images, we used images from the COCO 2017 (Lin et al., 2014) training set that are larger than 512×512, and downsampled each by a random factor in [0.6, 1] to remove potential compression artifacts from JPEG. We trained image compression models from (Minnen et al., 2018; Minnen & Singh, 2020) on our dataset and were able to closely reproduce their performance.

**Upper Bound Experiments** Instead of working with variational distributions over the set of pixel values $\mathcal{Y} = \mathcal{X} = \{0, 1, ..., 255\}^n$, which would require specialized tools for representing and optimizing high-dimensional discrete distributions (Salimans et al., 2017; Hoogeboom et al., 2019; Maddison et al., 2016; Jang et al., 2016), here we parameterize the variational distributions in Euclidean latent spaces for simplicity (see Sec. 3).

We applied our upper bound algorithm based on two VAE architectures:

1. The learned image compression model from Minnen & Singh (2020) [6]. This corresponds to a hierarchical VAE with two layers of latent variables, with the lower level latents modeled

---

[6] https://github.com/tensorflow/compression/blob/f9edc949fa58381ffafa5aa8cb37dc8c3ce50e7f/models/ms2020.py

by a channel-wise autoregressive prior conditioned on the top-level latents. We adapt it for our purpose, by using factorized Gaussians for the variational posterior distributions $Q_{Z|X}$ (instead of factorized uniform distributions used to simulate rounding in the original model), and no longer convolving the variational prior $Q_Z$ with a uniform distribution as in the original model.

2. The ResNet-VAE from our experiment on GAN images. Following learned image compression models, we maintain the convolutional structure of the topmost latent tensor (rather than flattening it in the GAN experiments) in order to make the model applicable to images of variable sizes. Instead of a MAF, we model the topmost latent tensor by a deep factorized (hyper-)prior $Q_{Z_0}$ as proposed by Ballé et al. (2018).

For the $\beta$-VAE based on Minnen & Singh (2020), $\dim(\mathcal{Z}) \approx 0.28 \dim(\mathcal{X})$; and for our $\beta$-ResNet-VAE, $\dim(\mathcal{Z}) \approx 0.66 \dim(\mathcal{X})$. We did not observe improved results by simply increasing the number of latent channels/dimensions in the former model. We follow the same model training practice as reported by Minnen & Singh (2020), training both models to around 5 million gradient steps.

In our problem setup, we consider the reproduction alphabet to be the discrete set of pixel values $\mathcal{X} = \{0, 1, ..., 255\}^n$, so the decoder network $\omega$ needs to discretize the output of the final convolutional layer (in `Tensorflow`, this can be implemented by a `saturate_cast` to `uint8`). However, since the discretization operation has no useful gradient, we skip it during training, and only apply it for evaluation.

The operational quality-rate curves of various compression methods are taken from the `tensorflow-compression`[7] and `compressAI`[8] libraries. The results of Minnen & Singh (2020) represent the current state-of-the-art of neural image compression, to the best of our knowledge.

**Lower Bound Experiments**   When running our lower bound algorithm on $256 \times 256$ crops of natural images, we observed similar behavior as on high-dimension Gaussians, with the loss $\tilde{\ell}_k$ quickly plateauing to a value around $\log k$. This issue persisted when we tried different $u$ model architectures. See a discussion on this phenomenon in Section A.5.2.

A.6   MEASURES OF VARIABILITY

As alluded to in Sec. 3, the proposed R-D bounds are estimated as empirical averages from samples, which only equal the true quantities in expectation. Before we characterize the variability of the bound estimates, we first define the exact form of the estimators used.

**Upper Bound Estimator**   For the proposed upper bound, consider the variational problem solved by the Lagrangian Eq. 3, and define the random variables from the rate and distortion terms of the Lagrangian:
$$\mathcal{R}(Z, X) := \log q(Z|X) - \log q(Z),$$
and
$$\mathcal{D}(Z, X) := \rho(X, \omega(Z)).$$
Above, $X$ and $Z$ follow the joint distribution $P_X Q_{Z|X}$, $\omega$ is the decoder function, and $q(z|x)$ and $q(z)$ are the Lebesgue density functions of absolutely continuous $Q_{Z|X=x}$ and $Q_Z$ variational distributions (we only worked with absolutely continuous variational distributions for simplicity). Then it is clear that the expectations of the two random variables equal the rate and distortion terms, i.e.,
$$\mathbb{E}[\mathcal{R}(Z, X)] = \mathbb{E}_{x \sim P_X}[KL(Q_{Z|X=x}\|Q_Z)],$$
and
$$\mathbb{E}[\mathcal{D}(Z, X)] = \mathbb{E}_{P_X Q_{Z|X}}[\rho(X, \omega(Z))].$$
Recall for any given decoder and variational distributions, the above rate and distortion terms define a point $(\mathbb{E}[\mathcal{D}], \mathbb{E}[\mathcal{R}])$ that lies on an upper bound $R_U(D)$ of the source $R(D)$ curve. Given $m$ i.i.d.

---

[7] https://github.com/tensorflow/compression/tree/8692f3c1938b18f123dbd6e302503a23ce75330c/results/image_compression

[8] https://github.com/InterDigitalInc/CompressAI/tree/baca8e9ff070c9f712bf4206b8f2da942a0e3dfe/results

| $\lambda$ | $\mu_{\mathcal{R}}$ | $s_{\mathcal{R}}$ | $\mathbb{E}[\mathcal{R}]$ CI | $\mu_{\mathcal{D}}$ | $s_{\mathcal{D}}$ | $\mathbb{E}[\mathcal{D}]$ CI |
|---|---|---|---|---|---|---|
| 300 | 0.278 | 0.8 | (0.15, 0.41) | 0.006 | 0.00287 | (0.00553, 0.00647) |
| 500 | 1.25 | 1.27 | (1.04, 1.46) | 0.00351 | 0.00207 | (0.00317, 0.00385) |
| 1e+03 | 2.69 | 1.21 | (2.49, 2.89) | 0.00157 | 0.00103 | (0.0014, 0.00174) |
| 4e+03 | 4.77 | 1.25 | (4.56, 4.97) | 0.000352 | 0.000285 | (0.000305, 0.000399) |
| 8e+03 | 5.65 | 1.31 | (5.44, 5.87) | 0.000196 | 0.000152 | (0.000171, 0.000221) |
| 1.6e+04 | 6.43 | 1.32 | (6.21, 6.64) | 0.000126 | 9.68e-05 | (0.00011, 0.000142) |

Table 1: Statistics for the estimated R-D upper bound on GAN-generated images with $d = 2$, computed with $m = 100$ samples of $(\mathcal{R}, \mathcal{D})$. "CI" denotes 95% large-sample confidence interval.

| $\lambda$ | $\mu_{\mathcal{R}}$ | $s_{\mathcal{R}}$ | $\mathbb{E}[\mathcal{R}]$ CI | $\mu_{\mathcal{D}}$ | $s_{\mathcal{D}}$ | $\mathbb{E}[\mathcal{D}]$ CI |
|---|---|---|---|---|---|---|
| 300 | 0.265 | 0.689 | (0.15, 0.38) | 0.00978 | 0.00376 | (0.00916, 0.0104) |
| 500 | 1.44 | 1.24 | (1.23, 1.64) | 0.00693 | 0.00284 | (0.00646, 0.00739) |
| 1e+03 | 4.08 | 1.73 | (3.80, 4.36) | 0.00332 | 0.00152 | (0.00307, 0.00357) |
| 2e+03 | 6.59 | 1.88 | (6.28, 6.90) | 0.00151 | 0.000752 | (0.00139, 0.00164) |
| 4e+03 | 8.66 | 1.89 | (8.35, 8.97) | 0.000774 | 0.000389 | (0.00071, 0.000838) |
| 8e+03 | 10.4 | 1.89 | (10.06, 10.68) | 0.000476 | 0.000229 | (0.000438, 0.000514) |
| 1.6e+04 | 12 | 1.83 | (11.69, 12.29) | 0.000323 | 0.000147 | (0.000299, 0.000347) |

Table 2: Statistics for the estimated R-D upper bound on GAN-generated images with $d = 4$, computed with $m = 100$ samples of $(\mathcal{R}, \mathcal{D})$. "CI" denotes 95% large-sample confidence interval.

pairs of $(Z_1, X_1), (Z_2, X_2), ..., (Z_m, X_m) \sim P_X Q_{Z|X}$, we estimate such a point by $(\mu_{\mathcal{D}}, \mu_{\mathcal{R}})$, using the usual sample mean estimators $\mu_{\mathcal{R}} := \frac{1}{m} \sum_{j=1}^{m} \mathcal{R}(Z_j, X_j)$ and $\mu_{\mathcal{D}} := \frac{1}{m} \sum_{j=1}^{m} \mathcal{D}(Z_j, X_j)$.

**Lower Bound Estimator** For the proposed lower bound, given i.i.d. $X_1, ..., X_k \sim P_X$, define the random variable

$$\xi := -\frac{1}{k} \sum_{i}^{k} \log u(X_i) - \frac{C_k}{\alpha} - \log \alpha + 1,$$

where the various quantities are defined as in Sec. 4. Then it is clear that its expectation equals the lower bound objective Eq. 8, i.e., $\mathbb{E}[\xi] = \tilde{\ell}_k := \mathbb{E}[-\log u(X)] - \mathbb{E}[C_k]/\alpha - \log \alpha + 1$. Given $m$ i.i.d. samples of $\xi$, we form the usual sample mean estimator $\mu_{\xi} := \frac{1}{m} \sum_{j} \xi_j$, and form an R-D lower bound estimate by

$$\hat{R}_L(D) := -\lambda D + \mu_{\xi}.$$

In computing $\mu_{\xi}$, we fix the constant $\alpha$ to a sample mean estimate of $\mathbb{E}[C_k]$ computed separately (as in line 7 of Algorithm 1). As explained in Sec. A.4, we repeat this procedure with different $\lambda$, and our overall R-D lower bound is obtained by taking the point-wise maximum of such linear under-estimators over $\lambda$.

**Results** Below we give measures of variability of for the sandwich bound estimates obtained on GAN generated images (Section 6.3). We report detailed statistics in the following tables, and plot them in Figure 12. Our results on other experiments have comparable variability.

For the upper bound, given each trained $\beta$-VAE, we compute $m$ samples of $(\mathcal{R}, \mathcal{D})$, essentially passing $m$ samples of $X$ through the autoencoder and collect the rate and distortion values; then we report the sample mean ($\mu$), sample standard deviation ($s$), and 95% large-sample confidence interval for the population mean (the true expected value) for $\mathcal{R}$ and $\mathcal{D}$, respectively, in Tables 1 and 2.

For the lower bound, given each trained $\log u$ model, we compute $m$ samples of $\xi$, then report the sample mean ($\mu_{\xi}$), sample standard deviation ($s_{\xi}$), and 90% large-sample confidence lower bound for the population mean (the true expected value $\mathbb{E}[\xi] = \tilde{\ell}_k$) in Tables 3 and 4.

| $\lambda$ | $\mu_\xi$ | $s_\xi$ | $\mathbb{E}[\xi]$ LCB |
|---|---|---|---|
| 99.63 | 0.74 | 0.01 | 0.74 |
| 199.90 | 1.46 | 0.01 | 1.45 |
| 299.15 | 2.04 | 0.02 | 2.03 |
| 499.69 | 2.78 | 0.02 | 2.78 |
| 999.01 | 3.84 | 0.03 | 3.83 |
| 1999.70 | 4.58 | 0.05 | 4.57 |
| 2024.00 | 4.60 | 0.06 | 4.58 |
| 2999.30 | 4.97 | 0.05 | 4.95 |
| 3036.80 | 4.96 | 0.08 | 4.93 |
| 4000.00 | 5.18 | 0.05 | 5.17 |

Table 3: Statistics for the estimated R-D lower bound on $d = 2$ GAN-generated images, computed with $m = 30$ samples of $\xi$. "LCB" denotes 90% large-sample lower confidence bound.

Figure 12: Zoomed-in version of Figure 3-Middle-Bottom (GAN-generated images with intrinsic dimension $d = 4$), showing the sample mean estimates of R-D upper bound points $(\mu_\mathcal{D}, \mu_\mathcal{R})$ bracketed by 95% confidence intervals (**red**), as well as lower bound estimates based on the sample mean $\mu_\xi$ (**maroon**, dashed) and its 90% confidence lower bound (**pink** line, beneath **maroon**).

| $\lambda$ | $\mu_\xi$ | $s_\xi$ | $\mathbb{E}[\xi]$ LCB |
|---|---|---|---|
| 99.63 | 1.09 | 0.01 | 1.09 |
| 199.90 | 2.18 | 0.02 | 2.17 |
| 299.15 | 3.08 | 0.02 | 3.07 |
| 499.69 | 4.44 | 0.04 | 4.43 |
| 999.01 | 5.97 | 0.06 | 5.95 |
| 1999.70 | 6.69 | 0.08 | 6.67 |
| 2024.00 | 6.69 | 0.07 | 6.66 |
| 2999.30 | 6.87 | 0.08 | 6.84 |
| 3036.80 | 6.85 | 0.09 | 6.83 |
| 4000.00 | 6.93 | 0.08 | 6.91 |

Table 4: Statistics for the estimated R-D lower bound on $d = 4$ GAN-generated images, computed with $m = 30$ samples of $\xi$. "LCB" denotes 90% large-sample lower confidence bound.

