# OpenReview forum: "Towards Empirical Sandwich Bounds on the Rate-Distortion Function"
_ICLR.cc/2022/Conference — ICLR 2022 Poster_

### Official Review · Reviewer_9Pwc · 2021-10-25

**Correctness:** 2
**Technical Novelty And Significance:** 2
**Empirical Novelty And Significance:** 1
**Recommendation:** 3
**Confidence:** 4

**Main Review:**

I cannot find any strength. I will outline major weakness, not covering all the waeknesses (as all pages are weak).

(1) The paper claims to propose upper and lower bounds on the R-D function of unknown memoryless information source. There are always nonzero probability of the "upper bound" being lower than the true R-D function, because the data samples are probabilistic. The standard tool for handling such a situation is one-sided confidence interval (see any undergraduate textbook on frequentiest statistics).
There is no reason for not using the standard statistical framework.
If one wants to use the Baysian counterpart of the interval estimation, one must have a prior probability of the unknown memoryless information source, but such a prior probability is not provided in the manuscript.

(2) The paper gives the big claim in the first page that "bounding the R-D function of a general (i.e. discrete, absolutely continuous or neither), unknown memoryless source". There are several subtleties, for example: The expectations of random variables do not
always exist, for example the Cauchy distribution. What is the definition of the R-D function for such probability distributions?
The expectation is used in almost every displayed equations of the manuscript. What if the expectation is undefined, like the case of the
Cauchy distribution? If the authors assume the existence of expectation in every mathematical expression of the manuscript,
what kind of the memoryless information source can be handled by the proposal? It is clear that the discrete and Gaussian sources
are OK, but they can be handled by the Arimoto-Bluhat algorithm in 1970s, after estimating the probability distribution by standard
statistical methods.

(3-1) The manuscript uses mathematical theorems that have not been proved for "general" memoryless information source.
For example, Lemma A.1 cites Csiszar and Gray, which proved Lemma A.1 for discrete and Gaussian sources.
The manuscript must prove Lemma A.1 for the class of information source that are considered.

(3-2) Similar for Theorem A.2. Please provide a proof of Theorem A.2 for the "general" information source.

(4) With any conficence interval estimation algorithm, there always exists a positive probability of the event that the true parameter does not belong
to the confidence interval generated by the algorithm with data samples obtained. The confidence level is one minus such a "failure"
probability. Thus, every practical interval estimation algorithm accompanies with (desirably theoretical) derivation of the confidence level.
The manuscript does provide confidence levels that can handle "general" memoryless information source.
It is acceptable that confidence levels of the proposed "upper" and "lower" bounds are also algorithmically produced given
data samples.

(5) Confidence levels should be evaluated with the finite number n of data samples, otherwise one cannot know what is the probabilities
of the proposed "lower" and "upper" bounds fail in a practical situation.

(6) If the authors want to retain their claim that their proposed algorithm can handle any probability distribution, please provide
the results of running their proposed lower and upper bounds algorithms on the Cauchy distribution as defined at
https://en.wikipedia.org/wiki/Cauchy_distribution

(7) Related to (6), it is unclear what kind of random variables/probability distributions can be handled by the proposed algorithms.
It must be clearly stated. Experiments are only conducted for the Gaussian distribution and so-called the "banana" source.
Are the proposed algorithms can be applied to other classes of probability distributions? If the authors claim wider applicability
of their proposed algorithm, it must be supported by experimental or theoretical evidences. If the proposals are only applicable
to a limited class of distributions, it is unsuitable for ICLR.

**Summary Of The Paper:**

The paper claims to propose "upper bound" and "lower bound" on the R-D function in lossy data compression.

**Summary Of The Review:**

Mathematical reasoning in the paper is doubtful, and what kind of situations can be handled by the proposed
algorithm is unclear, see "main review" for the detail.

---

> ### Author Response · Authors · 2021-11-23
> **We appreciate your insight, and hope you reevaluate based on our response and improved manuscript (part 1).**
>
> We thank the reviewer for the feedback, and hope to clear up any misunderstanding. To summarize, we see there are three main concerns:
> 1. *Variability of the results should be reported* (weaknesses (1), (4), (5)): Agreed, and please see new Appendix section A.6.
> 2. *The theorems A.1 are A.2 have not be proven for our setting* (weakness (3)): This is factually incorrect; the references gave proofs for the general setting (standard Borel alphabets) we consider.  To avoid future confusion, in Sec. 2 we now point readers to the "general setting of the paper, including the technical definitions, in Appendix A.1".
>
> 3. *The method cannot handle any probability distribution, such as the Cauchy* (weaknesses (2), (6), (7)): We believe there was potentially miscommunication, as we did not claim to be able to handle any arbitrary distribution. To prevent this in the future, we added new text at the end of Sec 2 requiring the existence of expectations w.r.t. the data distribution. This is a fundamental assumption of machine learning methods based on empirical risk minimization, such as maximum likelihood estimation.
>
> We address the first two main concerns in detail in this part 1 of our response, and address the third main concern in part 2.
>
> > (1) There are always nonzero probability of the "upper bound" being lower than the true R-D function, because the data samples are probabilistic. The standard tool for handling such a situation is one-sided confidence interval...
>
> > (4) ...every practical interval estimation algorithm accompanies with (desirably theoretical) derivation of the confidence level...
>
> > (5) Confidence levels should be evaluated with the finite number n of data samples...
>
> We agree with the assessment. In Appendix section A.6, we now explicitly derive/define our estimators used in the reported results. For both upper and lower bounds, our estimators are sample mean estimators of the population mean (which are proven upper and lower bounds on R-D), and we report the large-sample confidence bounds on them, as well as the number of samples used (m). The variability of the results do not appear significant in our plots, hence does not affect our findings. We are happy to provide additional statistics upon request.
>
> > (3-1) The manuscript uses mathematical theorems that have not been proved for "general" memoryless information source.
>
> > (3-2) Similar for Theorem A.2. Please provide a proof of Theorem A.2..
>
> The proofs are contained in the provided references, pp. 252 - 263 of [Gray, 2011], and [Csiszar, 1974], please double check. Our notion of a general source is clarified in the first paragraph of our appendix, i.e., requiring only that the alphabets are standard Borel spaces (the same as in [Harrison and Kontoyiannis 2008]). Csiszar's result is in fact more general, allowing the alphabets to be arbitrary measurable spaces (section 1 of [Csiszar, 1974]). To clarify this, we now state at the beginning of our Background section that "We formally describe the general setting of the paper, including the technical definitions, in Appendix A.1".

---

> > ### Comment · Reviewer_9Pwc · 2021-12-02
> > **My comment (3-1) and (3-2) were incorrect.**
> >
> > I agree to the authors' response to my comments (3-1) and (3-2). I withdraw comments (3-1) and (3-2).

---

> > > ### Comment · Reviewer_9Pwc · 2021-12-02
> > > **I upgraded my recommendation.**
> > >
> > > I edited the score part of the original report and upgraded some of scores (to positive directions) based on the authors' responses.

---

> ### Author Response · Authors · 2021-11-23
> **We appreciate your insight, and hope you reevaluate based on our response and improved manuscript (part 2).**
>
> We now address the reviewer's concern about the applicability of our method.
>
> > (2)  It is clear that the discrete and Gaussian sources are OK, but they can be handled by the Arimoto-Bluhat algorithm in 1970s, after estimating the probability distribution by standard statistical methods.
>
> Could you please elaborate on the feasibility of this approach, e.g., on a 1000-dimensional Gaussian as we considered in the paper? As far as we know, the BA algorithm does not handle a continuous alphabet, so the Gaussian must be discretized before any attempt to estimate it. As we mentioned in the introduction and section 3, the number of discretization bins increases exponentially in the data dimensionality, and quickly becomes intractable in more than 2 or 3 dimensions. Our experiments on high-dimension Gaussian, projections of the banana source, and 128x128 GAN-generated images clearly go beyond this setting,
>
> > (2) The paper gives the big claim ... that "bounding the R-D function of a general (i.e. discrete, absolutely continuous or neither), unknown memoryless source". There are several subtleties, for example: The expectations of random variables do not always exist.
>
> > (6) If the authors want to retain their claim that their proposed algorithm can handle any probability distribution, please provide the results ... on the Cauchy...
>
> We never claimed that our method can handle *any* arbitrary distribution.  Our claim of applicability to a *general* source (“continuous, discrete, or neither”, as we wrote) is to contrast with the prior art, i.e., the Blahut-Arimoto algorithm, which can only handle a discrete source. The word “general” is conventionally used in this context to refer to an alphabet being a general space, as in prior work in information theory [Csiszar, 1974; Gray, 2011; Kostina, 2016].  See also part 1 of our response for our exact definition of a general source (alphabets being standard Borel spaces).
>
> > (7) Related to (6), it is unclear what kind of random variables/probability distributions can be handled by the proposed algorithms. It must be clearly stated.
>
> Our original manuscript already stated the conditions under which the theorems (and hence our method) apply, and we encourage the reviewer to take a closer look.  As further evidence of applicability, see also our new experiment on realistic GAN-generated images.
> The theorems on the variational characterization of $R(D)$ are all prefaced with the basic condition that $D > D_{min}$, where $D_{min} = \inf \{ D’: R(D’) < \infty \}$ is the smallest distortion tolerance such that $R(D)$  is finite. i.e., the distortion level $D$ of interest must be such that $R(D)$ is finite for the results to be interesting.
> In pathological cases where the expectations may not exist, e.g., if the distortion $E[\rho(X, \hat X)] = \infty$ for all $Q_{\hat X| X}$, then $R(D) = \infty$; the theorems still hold vacuously, but no useful results can be obtained.  The lower bound theorem has an additional technical requirement (#2 in theorem A.2), essentially ensuring that it’s possible to quantize the source to finite distortion. Note that all the requirements hold automatically in the case of discrete data, since the R-D function is always finite and upper bounded by the data entropy. This ensures the applicability of our method to most digital data, such as images and audio, which are stored with finite bit precision.
>
> The issue of non-existing expectations with respect to the data distribution, such as with the Cauchy, poses a serious issue for any machine learning method based on empirical risk minimization, e.g., maximum likelihood estimation, and is not a specific limitation of our method. This is because standard machine learning relies on the concentration of measure phenomenon, such as the Laws of Large Numbers, to ensure empirical risk generalizes to true risk. For this to even make sense, the true risk (“test loss”, “generalization error”), defined as an expectation with respect to the data source distribution, needs to exist. We believe the pathological cases (such as the Cauchy) are rare in applications, given the empirical success of machine learning on real world problems.
>
> --------------
> References
>
> [Csiszar 1974] Imre Csiszar. On an extremum problem of information theory. Studia Scientiarum Mathematicarum Hungarica, 9, 01 1974.
>
> [Gray 2011] Robert M Gray. Entropy and information theory. pp. 252 - 263. Springer Science & Business Media, 2011
>
> [Kostina 2016] Victoria Kostina. When is Shannon's lower bound tight at finite blocklength? In 2016 54th Annual Allerton Conference on Communication, Control, and Computing (Allerton), pp. 982–989. IEEE, 2016
>
> [Harrison and Kontoyiannis 2008] Matthew T. Harrison and Ioannis Kontoyiannis. Estimation of the rate–distortion function. IEEE
> Transactions on Information Theory, 54(8):3757–3762, Aug 2008. ISSN 0018-9448. doi: 10.1109/
> tit.2008.926387. URL http://dx.doi.org/10.1109/TIT.2008.926387.

---

### Official Review · Reviewer_T3TM · 2021-10-29

**Correctness:** 4
**Technical Novelty And Significance:** 3
**Empirical Novelty And Significance:** 2
**Recommendation:** 6
**Confidence:** 3

**Main Review:**

Novelty: The classical Blahut-Arimoto algorithm requires the restrictive assumption of a discrete source as well as a known source distribution. This paper aims to develop a technique for unknown sources with possible continuous support. While there has been some work on extending this in the past (Riegler et al, 2018; Harrison & Kontoyiannis, 2008; Gibson, 2017), none have tackled the problem to the breadth of this paper. The paper is well written and easy to follow.

Significance: On the technical side the main contribution of the paper is on computing lower bound for rate-distortion function, as the upper bound is a rather natural application of the ELBO analysis in VAEs. While the experiments on the Gaussian and Banana sources give credence to the tightness of the bounds, these are rather simple examples. In particular, the variational distribution used for the experiments is Gaussian and contains the source distribution in its family. Although the banana source is a nonlinear transform of a Gaussian source, by visual inspection the underlying inverse transform seems to be easily learnable by an encoder in principle.  Do the authors have examples of synthetic experiment in which the source distribution is significantly different from the variational/latent distributions (Q_{Z|X}, Q_Z) as well?

The results on image compression suggesting a possible 1 PSNR improvement seem to be significant.
As the authors have mentioned, it is unclear whether we can actually close this gap due to the nature of it being an overestimate. Furthermore, the lower bound computation, which seem to be the main technical innovation does not scale to such datasets. As a result the reviewer feels that the technical contribution of the paper in realistic image compression applications may be limited due to these reasons.



**Summary Of The Paper:**

This paper considers the problem of estimating the rate-distortion function R(D) for arbitrary sources with unknown distribution. An upper bound is established using a variational distribution and is estimated using an iterative coordinate descent algorithm inspired by the Blahut-Arimoto algorithm. A lower bound is established using a parameterization of the dual form of the rate-distortion function by Cisszar. These are then evaluated using neural network encoder/decoders on Gaussian and Banana sources, and tested on natural images suggesting that there is room for improved compression rates achieved by current state-of-the-art image compression algorithms.



**Summary Of The Review:**

Due to the weaknesses in the "significance" part of the main review I am marking the score as a 5.

---

> ### Author Response · Authors · 2021-11-23
> **Your review has helped strengthened this paper. We now include R-D sandwich bounds on 128x128 GAN-generated images.**
>
> > An upper bound is established using a variational distribution and is estimated using an iterative coordinate descent algorithm inspired by the Blahut-Arimoto algorithm.
>
> A minor nitpick in case there was misunderstanding: our upper algorithm performs *gradient* descent rather than *coordinate descent*, therefore avoids some of the difficulties of the Blahut-Arimoto algorithm.
>
> > This paper aims to develop a technique for unknown sources with possible continuous support. While there has been some work on extending this in the past (Riegler et al, 2018; Harrison & Kontoyiannis, 2008; Gibson, 2017), none have tackled the problem to the breadth of this paper. The paper is well written and easy to follow.
>
> Thank you for recognizing the scope of the problem we are tackling. We hope this work is a useful first step towards solving this problem, and helps bridge the gap between rate-distortion theory and practice.
>
> > Although the banana source is a nonlinear transform of a Gaussian source, by visual inspection the underlying inverse transform seems to be easily learnable by an encoder in principle.
>
> You have the right intuition; as in a VAE, the encoder learns to simplify the data into a representation that is easier to model in the latent space. However, we note that even if the nonlinear transform is analytically known (as in the banana experiments), the R-D function of the transformed source is still non-trivial to obtain. There’s no simple relation between the R-D function of a source $X$ and that of its transformation $f(X)$ for an arbitrary function $f$.
> (e.g., we do not even know the analytical form of the R-D function of the Gaussian with a full covariance matrix, which can be obtained by a simple linear transform of the standard Gaussian)
>
>
> > Do the authors have examples of synthetic experiment in which the source distribution is significantly different from the variational/latent distributions (Q_{Z|X}, Q_Z) as well?
>
> We agree that the banana source might seem simple to model by a VAE; nonetheless we still needed to use flexible normalizing flows (stacks of MAF and IAF) for the variational distributions to obtain a tight upper bound in Fig. 2.
>
> To remove any doubts, we applied our method on 128x128 GAN-generated images, and used our sandwich bounds to assess existing learned image compression methods.  The distribution of GAN generated images, such as the dog with various poses, body proportions, and facial features, lies on a complex manifold of $[0, 1]^n, n=128 \times 128 \times 3$ that can not be well captured by a Gaussian distribution, while we use factorized Gaussian variational distributions at each level of the hierarchical VAE.

---

### Official Review · Reviewer_zsSF · 2021-11-01

**Correctness:** 4
**Technical Novelty And Significance:** 3
**Empirical Novelty And Significance:** 3
**Recommendation:** 8
**Confidence:** 3

**Main Review:**

The paper is well-motivated and has valuable results for the lossy compression community. In more detail:

$\textbf{Strenghts:}$
- The paper is well-written and summarizes the prior work well for readers from both information theory and generative models backgrounds.

- The authors provide a tight upper bound on the RD function of arbitrary sources without limiting the sources to discrete data or restricting the dimensions. I believe this is a useful contribution to the lossy compression field. For instance, as the authors pointed out in the paper, we can assess the success of lossy compression algorithms by comparing their performance with tight RD bounds.

-  The authors use the dual formulation of the RD function to find a lower bound, which, to the best of my knowledge, is a novel approach.

- The experimental results on random Gaussian samples show the exactness of their upper bound (for this particular source). For the banana-shaped source and its high-dimensional projection, the authors show that upper and lower bounds are tighter than what was proposed in prior work. These two sets of experiments provide empirical support for the theoretical results in the paper.

- I find the experimental results on natural images interesting as the authors essentially set a (somewhat loose) limit for future work in lossy image compression without actually providing a compression algorithm.

$\textbf{Weaknesses:}$
- Despite the flexibility of the proposed method for finding upper bounds on RD function, the authors restrict the source to have low effective dimension with continuous reproduction alphabet. This makes the current lower bound not applicable to high-dimensional data such as images.

- The lower bound is evaluated only on a banana-shaped source. It would be nice to see the tightness of the upper and lower bounds on a few other sources.

- I could not access the code in the given link. The google drive folder seems empty.

- A few minor points:
    - Section 4, Dual Characterization of R(D) paragraph, "..., a variational lower would require..." misses the word "bound"
    - Section 6.1, last sentence of the second paragraph: optional - - > optimal
    - There are a few problems with the Appendix. The text for the last section A.5.3 Natural Images seems missing. The figure number in the last paragraph is also missing.

**Summary Of The Paper:**

This paper aims to provide stronger upper and lower bounds for the RD function of arbitrary sources. The authors handle unknown sources by requiring only i.i.d. samples. Specifically, the authors derive an upper bound to the RD function using a $\beta$-VAE-like generative model, which has some similarities to the Blahut-Arimoto (BA) algorithm with two distinctions: 1) they do not restrict the source to be a low-dimensional discrete data, and 2) they use gradient descent to predict the parameters of the variational distributions $Q_{\hat{X}|X}$ and $Q_{\hat{X}}$. They also derive a lower bound to the RD function using the dual form of the RD function. Finally, the authors provide experimental results to show that the proposed upper bound is, in fact, exact for random Gaussian samples. For more complex data such as banana-shaped sources and their high-dimensional projections, they show that the proposed upper and lower bounds are tight. Furthermore, the proposed upper bounds on the RD function of natural images indicate that the state-of-the-art image compression methods can still be improved by at least one dB PSNR.

**Summary Of The Review:**

Although the proposed method has limitations, I find some of the contributions valuable.

---

> ### Author Response · Authors · 2021-11-23
> **Thank you for recognizing the significance of this work. Your feedback has helped us further improve it.**
>
> > I believe this is a useful contribution to the lossy compression field. For instance, as the authors pointed out in the paper, we can assess the success of lossy compression algorithms by comparing their performance with tight RD bounds.
>
> We indeed perform this assessment in our latest experiments in Sec. 6.2.  Although we didn’t have space to mention this, the R-D function is also useful for assessing the minimum channel capacity required by a combined source-channel coding system (see the Source-Channel Separation Theorem, e.g. in section 7.13 of Thomas and Cover).
>
> > the authors restrict the source to have low effective dimension with continuous reproduction alphabet. This makes the current lower bound not applicable to high-dimensional data such as images.
>
> > The lower bound is evaluated only on a banana-shaped source. It would be nice to see the tightness of the upper and lower bounds on a few other sources.
>
> Please see our latest sandwich bounds on realistic GAN-generated images, and comparisons to two popular learned image compression methods, in Sec. 6.2. Again, we show that the high dimension of the data (in this case, $n=128 \times 128\times 3$) does not necessarily pose a problem for our lower bound algorithm, consistent with our earlier finding. Our sandwich bounds allow us to assess the operational rate-distortion performance of learned image compression methods (e.g., the operational R-D curve of [Minnen et al., 2021] lies roughly within a factor of 2 above our estimated true R-D), and reveal the effectiveness of the learned compression approaches at exploiting low-dimension structures in the data. In particular, as the intrinsic dimension of the images decreases, the operational R-D performance of these methods show clear improvement, following the same trend as our estimated true R-D region (see Figure 3-Middle), while this is non-trivial to achieve with a traditional compression method like JPEG, whose performance on this problem is orders of magnitude worse.
>
> > I could not access the code in the given link.
>
> > A few minor points: ...
>
> Thanks for bringing these issues to our attention. They have been fixed now.

---

### Official Review · Reviewer_tH6q · 2021-11-02

**Correctness:** 4
**Technical Novelty And Significance:** 3
**Empirical Novelty And Significance:** Not applicable
**Recommendation:** 6
**Confidence:** 3

**Main Review:**

The paper proposes to use ML to establish lower and upper bounds on rate distortion for general sources, thus going beyond the Blahut-Arimoto algorithm (which assumes discrete sources with known PMF, and has complexity exponential in the dimension of the data). Some theoretical results are provided, extending prior results. Some numerical results are provided.

To construct the upper bound, the paper starts with the unconstrained variational objective of Blahut-Arimoto.  The distributions Q(\hat{x} | x) and Q(\hat{x}) are parametrized by the parameters of a NN; minimizing the objective function provides a point on the upper bound of the R(D) curve. A key insight is the observation that this objective function is similar to NELBO in \beta-VAE.  Based on this a projection to a lower-dimensional space is considered. The paper establishes R_\omega(D) \ge R(D), where \omega is the decoder function. While this seems straightforward, there is no discussion of the complexity in finding each R(D) point, and of course the required sweep over \lambda to find the entire UB curve. There is little discussion on the choice of \omega; in particular what choices are likely to lead to tighter bounds.
The establishment of the lower bound is less clear. The paper notes that this much harder than the UB case. Again there is little detail on the architecture for computing u_\theta(x). At the end of section 4, we are told that Appendix A has the detailed algorithm, but Appendix A says that an outline will be provided

Numerical section: insufficient details are provided so that it would not be possible for an informed reader to reproduce the results in the paper. . While some detail is provided for the Gaussian sources (results shown in Fig 2), very little detail is provided for the banana sources (Fig 3) or the natural images (Fig 4).

Section A.5.3 “Natural Images” is a blank section

Other:

What does “1 PSNR” (used in the abstract and several other places mean)? Does this mean 1 dB improvement in PSNR?

I assume a diagonal Gaussian process is a vector Gaussian process in which the components are independent so that the covariance matrix is diagonal?



**Summary Of The Paper:**

The paper proposes to use ML to establish lower and upper bounds on rate distortion for general sources, thus going beyond the Blahut-Arimoto algorithm (which assumes discrete sources with known PMF, and has complexity exponential in the dimension of the data). Some theoretical results are provided, extending prior results. Some numerical results are provided

**Summary Of The Review:**

The use of ML to address complex problems in information theory is appealing. The authors consider the problem of obtaining upper and lower bounds for the rate distortion curve for source distributions that may be continuous, discrete, or mixed, and may not be known (only data are available). Some theoretical results are provided. Two major issues: insufficient details about the learning architectures are provided, so that an informed reader will not be able to reproduce the results in the paper. Second, there is very little discussion of the complexity of the algorithm.

---

> ### Author Response · Authors · 2021-11-23
> **Thank you, we have greatly improved the appendix and included the details.**
>
> > Some theoretical results are provided, extending prior results.
>
> While our upper bound algorithm extends the Blahut-Arimoto algorithm, we believe the connection to beta-VAE is novel, in particular our insight that a suitable beta-VAE optimizes an upper bound of the source R-D function.  Our proposed lower bound estimator is also novel, and to our best knowledge, the first such numerical procedure applied to large-scale problems and yield non-trivial sandwich bounds (see, e.g., our latest experiment in Sec. 6.2 on GAN-generated images).
>
>
> > “While this seems straightforward, there is no discussion of the complexity in finding each R(D) point, and of course the required sweep over \lambda to find the entire UB curve.
>
> The upper bound algorithm is indeed straightforward -- producing each R-D point has the same complexity as training a beta-VAE and evaluating it on samples. We now include a detailed lower bound algorithm and discuss the computation required (and ways to speed it up) in section A.4. We discuss the time and hardware used to train the models in section A.5.1., with the larger beta-VAE models taking the longest to train (a couple of days, comparable to image compression models such as [Minnen et al. 2020]).
>
> To reduce the computation when sweeping out R-D bounds, we experimented with “chaining” the model training, where we first train an upper bound model with the highest $\lambda$, then fine-tune it with a smaller $\lambda$ (we do the opposite for the lower bound: we start with a small $\lambda$ and tune it up). The fine-tuning typically requires significantly less computation than training from scratch, and gives equally good if not better results.
>
> > There is little discussion on the choice of \omega; in particular what choices are likely to lead to tighter bounds.
>
> In this work, \omega is simply the decoder portion of an autoencoder, which is usually a convolutional network with upsampled convolutions for the image experiments, and a MLP in all other experiments. In general, \omega should be a flexible function, e.g., a neural network, trained on the Lagrangian objective. We cannot say much on the exact neural architecture of \omega, as it depends on the problem setting, but at a high level it should be rich enough to cover various reconstruction points in the $\hat{\mathcal{X}}$ space to stay within the distortion tolerance $D$ for $R(D)$. We capture this intuition in theorem A.3, which points to bijectivity as a desirable criterion for the decoder (ensuring $R_\omega(D)=R(D)$), lending some theoretical support to the practice of replacing (non-invertible) upsampling with (invertible) pixel-shuffling operation in the decoders of image compression models (see, e.g., [Theis et al., 2017; Cheng et al., 2020]).
>
> > Again there is little detail on the architecture for computing u_\theta(x). At the end of section 4, we are told that Appendix A has the detailed algorithm, but Appendix A says that an outline will be provided”
>
> > Numerical section: insufficient details are provided ..
> > Section A.5.3 “Natural Images” is a blank section
>
> Thank you for bringing these issues to our attention. We now include the relevant discussions in Appendix sections A.4 and A.5.
>
>
> >What does “1 PSNR” (used in the abstract and several other places mean)? Does this mean 1 dB improvement in PSNR?
> >I assume a diagonal Gaussian process is a vector Gaussian process in which the components are independent so that the covariance matrix is diagonal?
>
> You’re correct on both accounts; sorry about the confusion. We’ve fixed these two wording issues in the revised manuscript.
>
> --------
> References
>
> [Minnen et al. 2020]  D. Minnen and S. Singh. Channel-wise autoregressive entropy models for learned image compression. In IEEE International Conference on Image Processing (ICIP), 2020.
>
>
> [Theis et al. 2017] L. Theis, W. Shi, A. Cunningham, and F. Huszar. Lossy Image Compression with Compressive Autoencoders. In International Conference on Learning Representations, 2017.
>
> [Cheng et al. 2020] Zhengxue Cheng, Heming Sun, Masaru Takeuchi, and Jiro Katto. Learned image compression with discretized gaussian mixture likelihoods and attention modules.
> arXiv preprint arXiv:2001.01568, 2020

---

### Author Response · Authors · 2021-11-23
**General remarks, part 1**

We thank the reviewers for their constructive feedback.
Overall the reviewers found the use of ML to address complex problems in information theory to be appealing (Reviewer tH6q), our work well-motivated (Reviewer zsSF), and the results valuable for the lossy compression community (Reviewer zsSF).
The reviewers were mainly concerned with the experimental results.  We have revised the manuscript extensively to address these concerns:

1. **Applicability of our proposed sandwich bounds to real-world data** (Reviewers tH6q, zsSF): we included new experiments establishing sandwich bounds on 128x128 realistic GAN-generated images with controlled latent dimension, confirming our earlier finding that our lower bound remains tight on high-dimension data with sufficiently low intrinsic dimension. This already allows us to assess the compression performance of popular learned image compression methods on specialized content (e.g., the operational R-D curve of [Minnen et al., 2020] lies roughly within a factor of two above our estimated true R-D), and reveals their effectiveness at exploiting low-dimension structure of the data.
2. **Experimental details and analysis of the complexity of the algorithms** (Reviewer T3TM): we updated the appendix extensively to include these details, in sections A.4, A.5.

3. **Variability of the reported results** (Reviewer 9Pwc): we now include example variability measures of our results, including sample variance and confidence intervals, in section A.6. In all of our results, the variability appears insignificant on the scale of the plots, hence does not affect our overall empirical findings. We are happy to provide additional results and statistics upon request.

One reviewer (9Pwc) in particular also questioned the mathematical foundation of our method, so we address them here at a high level, hoping to assure all reviewers of the soundness of our method.

Specifically, the reviewer has made the **factually incorrect** claim that the theorems A.1 and A.2, on which our methods are based, have not been proven for our problem setting. The opposite is true, and we refer interested readers to pp. 252 - 263 of [Gray, 2011] and the seminal paper by [Csiszar, 1974], which proved these results in the general setting we considered, i.e., the alphabets being standard Borel measure spaces. Csiszar in fact proved the results for the even broader setting of the alphabets being “arbitrary measurable spaces” (section 1 of [Csiszar, 1974]).


The reviewer also pointed out (correctly) that our method may not handle any arbitrary distribution, bringing out the Cauchy distribution as an example, and questioned the applicability of our method.

First, we made no claim of being able to compute the R(D) of *any* distribution in this paper. Our claim of applicability to a *general* source (“continuous, discrete, or neither”, as we wrote) is to contrast with the prior de facto method, i.e., the Blahut-Arimoto algorithm, which can only handle a discrete source. The word “general” is conventionally used in this context to refer to an alphabet being a general measurable space, as in prior work in information theory [Csiszar, 1974; Gray, 2011; Kostina, 2016].

Second, we clearly stated the conditions under which the theorems (and therefore our methods) apply, with a basic requirement that $R(D)$ is finite for the distortion level $D$ of interest. In case the expectations defining $R(D)$ are infinite, then $R(D) = \infty$, and no interesting results can be obtained. To avoid this vacuous case, we’ve now added a sentence at the end of section 2, supposing that “various expectations w.r.t. the data distribution exist”.  Note that these issues go away in the case of discrete data, since $R(D)\leq H[X]$, i.e., the R-D function is always upper bounded by the data entropy and therefore finite. This ensures the applicability of our method to most digital data, such as images and audio, which are represented with finite bit precision.

Third, we remark that the issue of non-existing expectations with respect to the data distribution, such as with a Cauchy distribution (for which Laws of Large Numbers do not hold), poses a serious issue for any machine learning method based on empirical risk minimization, e.g., maximum likelihood estimation, and is not a limitation specific to our method. For example, the expected risk of fitting a Gaussian density model to an unknown data distribution may be undefined, if the data turns out to be Cauchy-distributed, even though the empirical risk can always be (and is routinely) minimized in practice.  Such cases do not reflect the typical setting of machine learning, given the empirical success of the field and the ability of various models to generalize.

---

### Author Response · Authors · 2021-11-23
**General remarks, part 2**

We would like to re-emphasize the significance of our contribution, which appears underappreciated. As acknowledged by Reviewer T3TM,  “while there has been some work in the past”, “none have tackled the problem to the breadth of this paper”. In particular, R-D lower bounds have been recognized by the information theory community as “notoriously hard to obtain” [Riegler et al., 2018], and we are not aware of any existing numerical algorithm that produces non-trivial R-D lower bounds at the scale we experimented with. And since the only known baseline, the Blahut-Arimoto algorithm, fails to even apply in most of the settings considered, we believe our results already represent significant progress, and establish a valuable baseline for future work. While the general problem that we tackle here might not be solved any time soon, hopefully our work raises awareness of this research topic and represents a good step in bridging the gap between rate-distortion theory and practice.

-------------------------------------------------
References:

[Minnen et al. 2020] D. Minnen and S. Singh. Channel-wise autoregressive entropy models for learned image compression.
In IEEE International Conference on Image Processing (ICIP), 2020.

[Csiszar 1974] Imre Csiszar. On an extremum problem of information theory. Studia Scientiarum Mathematicarum Hungarica, 9, 01 1974.

[Gray 2011] Robert M Gray. Entropy and information theory. pp. 252 - 263. Springer Science & Business Media, 2011

[Kostina 2016] Victoria Kostina. When is Shannon's lower bound tight at finite blocklength? In 2016 54th Annual Allerton Conference on Communication, Control, and Computing (Allerton), pp. 982–989. IEEE, 2016

[Riegler et al., 2018] Erwin Riegler, Gunther Koliander, and Helmut Bolcskei. Rate-distortion theory for general sets and  measures. arXiv preprint arXiv:1804.08980, 2018.

---

### Decision · Program_Chairs · 2022-01-20

**Decision:**

Accept (Poster)

**Comment:**

This paper proposes an algorithmic approach to estimating upper and lower bounds of the rate-distortion (R-D) function of a data source on the basis of samples drawn from it. The proposed upper bound is based on the variational objective employed in the Blahut-Arimoto algorithm, whereas the proposed lower bound is based on the dual characterization of the R-D function. In both bounds neural networks trained with samples are utilized. Experimental results on four sources (Gaussian, banana-shaped, GAN-generated images, and natural images) are provided.

The four review scores were initially two positives and two negatives. Some reviewers evaluated positively the argument on the lower bound of the R-D function. On the other hand, one reviewer showed his/her concern about lack of the argument on statistical confidence of the obtained bounds. In response, the authors have addressed it in Section A.6 in the Supplementary Materials (SM) of the revised manuscript with the experiment using GAN-generated images (high-dimensional data with low intrinsic dimension), and the results are summarized in Tables 1-4 and Figure 10 in SM. The authors have in their revision also provided a specification for the range of the sources to which the proposal would be applicable. Description of the experiments on images, which was missing in the initial version as pointed out by some reviewers, has been added in the revised manuscript. Still, as some reviewers mentioned, the main weakness is that the proposal failed to demonstrate its usefulness to estimate the lower bound in settings where the data dimension is truly high, as in the experiment with natural images, where statistical confidence analysis was not conducted either. Three reviewers have revised their respective scores upward after the author response.

Despite some weaknesses I think that this paper provides a novel and interesting algorithmic approach to estimating the rate-distortion function. I would therefore like to recommend acceptance of this paper, and would like to encourage the authors to perform confidence analysis also for the experiments on natural images.